# Mechanism of mitoribosomal small subunit biogenesis and preinitiation

Yuzuru Itoh[1,9], Anas Khawaja[2,3,9], Ivan Laptev[4,5,6], Miriam Cipullo[2,3], Ilian Atanassov[7], Petr Sergiev[4,5,6,8], Joanna Rorbach[2,3✉] & Alexey Amunts[1✉]

Mitoribosomes are essential for the synthesis and maintenance of bioenergetic proteins. Here we use cryo-electron microscopy to determine a series of the small mitoribosomal subunit (SSU) intermediates in complex with auxiliary factors, revealing a sequential assembly mechanism. The methyltransferase TFB1M binds to partially unfolded rRNA h45 that is promoted by RBFA, while the mRNA channel is blocked. This enables binding of METTL15 that promotes further rRNA maturation and a large conformational change of RBFA. The new conformation allows initiation factor mtIF3 to already occupy the subunit interface during the assembly. Finally, the mitochondria-specific ribosomal protein mS37 (ref. [1]) outcompetes RBFA to complete the assembly with the SSU–mS37–mtIF3 complex[2] that proceeds towards mtIF2 binding and translation initiation. Our results explain how the action of step-specific factors modulate the dynamic assembly of the SSU, and adaptation of a unique protein, mS37, links the assembly to initiation to establish the catalytic human mitoribosome.

The human mitoribosome consists of the SSU that binds mRNA and tRNA to ensure accurate initiation and decoding, and the large subunit (LSU) that secures a nascent polypeptide arrival to the inner mitochondrial membrane via the OXA1L insertase[1–4]. Although recent structural studies have deepened our understanding of the mitoribosome structure, function and antibiotic binding[1–9], we still lack information on the assembly of the mitoribosomal components that would explain how they cooperatively reach the first functional state to initiate translation. In bacteria and the cytosol of eukaryotic cells, the assembly steps for the SSU include a series of alternating RNA conformations that rearrange in a stepwise manner into a native structure with a fully folded decoding region[10–14]. However, no structural data exist on any of the assembly steps of the SSU in mammalian mitochondria, and it remains unclear to what extent they are shared with the bacterial counterpart, and what are the tasks of mitochondria-specific components in ensuring a correct placement and folding of the rRNA. This is particularly important as defects in the assembly are linked to severe pathologies that manifest as a broad range of developmental disorders[15]. In this study, we determined sequential cryo-electron microscopy (cryo-EM) structures of stable assembly intermediates and initiation complexes at 2.8–3.2 Å resolution that involve six factors: RBFA, TFB1M, METTL15, mtIF3, mS37 and mtIF2 (Extended Data Figs. 1–4, Supplementary Tables 1 and 2, Supplementary Video 1 and Supplementary Figs. 1 and 2).

As several rRNA methyltransferases are required for the assembly, to stall the process, we first depleted the enzyme TRMT2B, responsible for the formation of 5-methyluridine[16,17] (Methods), which produced a viable cell line with stable rRNA and allowed to purify intermediate pre-SSU particles in different states. Judged by the proportion of the unfolded rRNA, the earliest intermediate pre-SSU-1 (3.2 Å resolution) is represented by partially disordered rRNA helices (h44 and h45) and two disordered linkers (h28–h44 and h44–h45) (Fig. 1a and Extended Data Figs. 1 and 2). It lacks the peripheral mitoribosomal protein mS37, and instead contains the assembly factors RBFA and dimethyladenosine transferase TFB1M (Fig. 1a and Extended Data Fig. 1). RBFA has a KH-like N-terminal domain (NTD), followed by a short C-terminal domain (CTD), which remained unresolved in the previous structures[10,18] (Fig. 1). Mitochondria-specific extensions nearly double the protein size (Extended Data Fig. 5), and we identified their specific functions. The association of RBFA leads to a vertical displacement of uS7m in the head by approximately 12 Å up, and the entire head is rotated approximately 10° towards the A site, compared to the mature SSU, thereby exposing the P site region (Extended Data Fig. 6a). Such an arrangement of RBFA occupies a position on the SSU, which is incompatible with mS37 (Fig. 1a, Extended Data Fig. 6b and Supplementary Video 2), and mitochondrial C-terminal extension (CTE) has a specific role by stretching over 60 Å through the entire mRNA channel to block its entry (Fig. 1a). RBFA further acts as a local rRNA scaffolder that promotes maturation by predisposing the hotspot modification region for TFB1M, and we detected two functional RBFA–rRNA contacts (Fig. 1b and Extended Data Fig. 6c): (1) binding of otherwise free rRNA 3′ end, which in the mature SSU is associated with the missing mS37; and (2) binding of rRNA h28 through a 30 Å-long surface, correlated with a dislocation of h28 by 12 Å towards the h44–h45 linker.

TFB1M is the methyltransferase that uses *S*-adenosylmethionine (SAM) to dimethylate two adjacent adenosine residues, A937 and A938 (mouse numbering, equivalent to human A1583 and A1584)

[1]Science for Life Laboratory, Department of Biochemistry and Biophysics, Stockholm University, Solna, Sweden. [2]Department of Medical Biochemistry and Biophysics, Karolinska Institute, Stockholm, Sweden. [3]Max Planck Institute for Biology of Ageing-Karolinska Institutet Laboratory, Karolinska Institutet, Stockholm, Sweden. [4]Center of Life Sciences, Skolkovo Institute of Science and Technology, Moscow, Russia. [5]Department of Chemistry, Lomonosov Moscow State University, Moscow, Russia. [6]Belozersky Institute of Physico-Chemical Biology, Lomonosov Moscow State University, Moscow, Russia. [7]Proteomics Core Facility, Max-Planck-Institute for Biology of Ageing, Cologne, Germany. [8]Institute of Functional Genomics, Lomonosov Moscow State University, Moscow, Russia. [9]These authors contributed equally: Yuzuru Itoh, Anas Khawaja. ✉e-mail: joanna.rorbach@ki.se; amunts@scilifelab.se

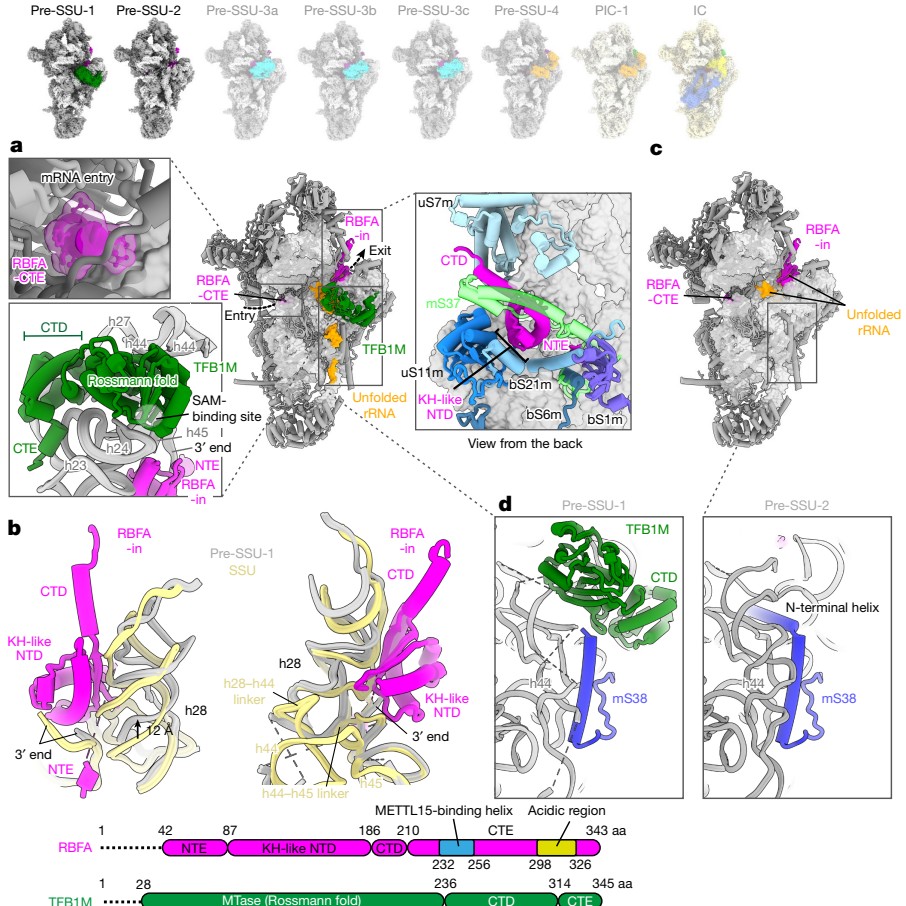

**Fig. 1 | Structures of pre-SSU-1 and pre-SSU-2 states. a**, Pre-SSU-1 with RBFA and TFB1M. Unfolded rRNA is in orange. RBFA-CTE (surface model) blocking the mRNA channel (upper left), TFB1M binding (bottom left), and superposition of pre-SSU-1 with mature SSU showing clashes of RBFA with mS37 (right) are displayed. **b**, Superposition of pre-SSU-1 with mature SSU rRNA shows an upwards displacement of h28 due to extended association with RFBA that also interacts with the rRNA 3′ end. **c**, Pre-SSU-2 with RBFA. **d**, Comparison of pre-SSU-1 and pre-SSU-2 shows that the mS38 N-terminal helix is disordered in pre-SSU-1. aa, amino acid; NTE, N-terminal extension.

(Supplementary Table 3), in the hairpin loop of h45. It has a CTD (residues 237–314), followed by a CTE (residues 315–345), forming an additional α-helix that together make a large interface of 2,077 Å² with the rRNA h23, h24, h27 and h45 (Fig. 1a and Extended Data Fig. 7a). The C-terminal α-helix, which is disordered in the crystal structures[19,20], further links TFB1M from h24 to the platform rRNA h23 (Extended Data Fig. 7a). In the SSU body, the N-terminal helix of the mitoribosomal protein mS38 is disordered, and its mature conformation would clash with TFB1M.

Comparison of our pre-SSU-1 and pre-SSU-2 (3.1 Å resolution) structures shows that TFB1M is missing in the latter, whereas rRNA is more ordered (Fig. 1a–c, Extended Data Figs. 1 and 7b and Supplementary Video 2). In addition, the N-terminal helix of mS38 is inserted into the rRNA groove, adopting its mostly mature conformation (Fig. 1d and Extended Data Fig. 7b). Thus, RBFA prepares pre-SSU-2 for a subsequent accommodation of the next assembly factor, and also prevents a premature association of mRNA and mS37.

In the same dataset, we also observed a class with a subset of particles with folded rRNA and the additional density that is different from TFB1M, suggesting a later sequential assembly intermediate. To explore how the SSU progresses through the mitoribosomal assembly pathway after maturation of TFB1M, we reasoned that an inhibition of the corresponding LSU assembly would allow the SSU to proceed towards later stages up to the subunit joining. Therefore, we knocked out the enzyme mitochondrial rRNA methyltransferase 3 (MRM3), which is responsible for a late assembly of the LSU[21], and collected cryo-EM data of the accumulated SSU and LSU particles (Extended Data Figs. 3, 4 and 8 and Supplementary Figs. 2 and 3). Approximately 7% of the SSU particles in this cryo-EM data showed identical structural features and the presence of the same additional density that were also detected in the particles from the TRMT2B-depleted cells, and the latter was identified as the $N^4$-methylcytidine methyltransferase METTL15 (Extended Data Figs. 1, 2a and 3a). METTL15 is the mitochondrial orthologue of the bacterial RsmH and is responsible for the methylation of C1486 in the SSU rRNA decoding centre[22–24]. We resolved this state (pre-SSU-3) at 2.8 Å resolution (Extended Data Fig. 1 and Supplementary Fig. 2). In pre-SSU-3, the rRNA h28, h44 and h45 are folded and all known rRNA modifications are observed (Fig. 2a). The improved resolution allowed us to reveal how mito-specific RBFA-CTE blocks mRNA binding: residues 278–298 fill the A site, residues 299–326 that are enriched with 13 Asp/Glu residues (see 'acidic region' in Fig. 1) is located at the mRNA entry and further interact with mS35 and mS39 to prevent recruitment of mRNA.

In coordination with METTL15 binding and rRNA maturation, RBFA adopts a large conformational change in pre-SSU-3 (from in to out) (Fig. 2b and Supplementary Video 2), which is enabled by a pivot point between the KH-like domain and CTD. The conformational change appears to be a unique feature of RBFA, which was not reported for bacterial counterparts[10,25]. METTL15 would clash with RBFA-in from pre-SSU-2 (Extended Data Fig. 9a), but forms direct interactions with RBFA-out through an α-helix (residues 232–256 in RBFA-CTE) (Fig. 2a), which is consistent with the report that METTL15 coprecipitates with RBFA[23]. Therefore, the association of METTL15 triggers

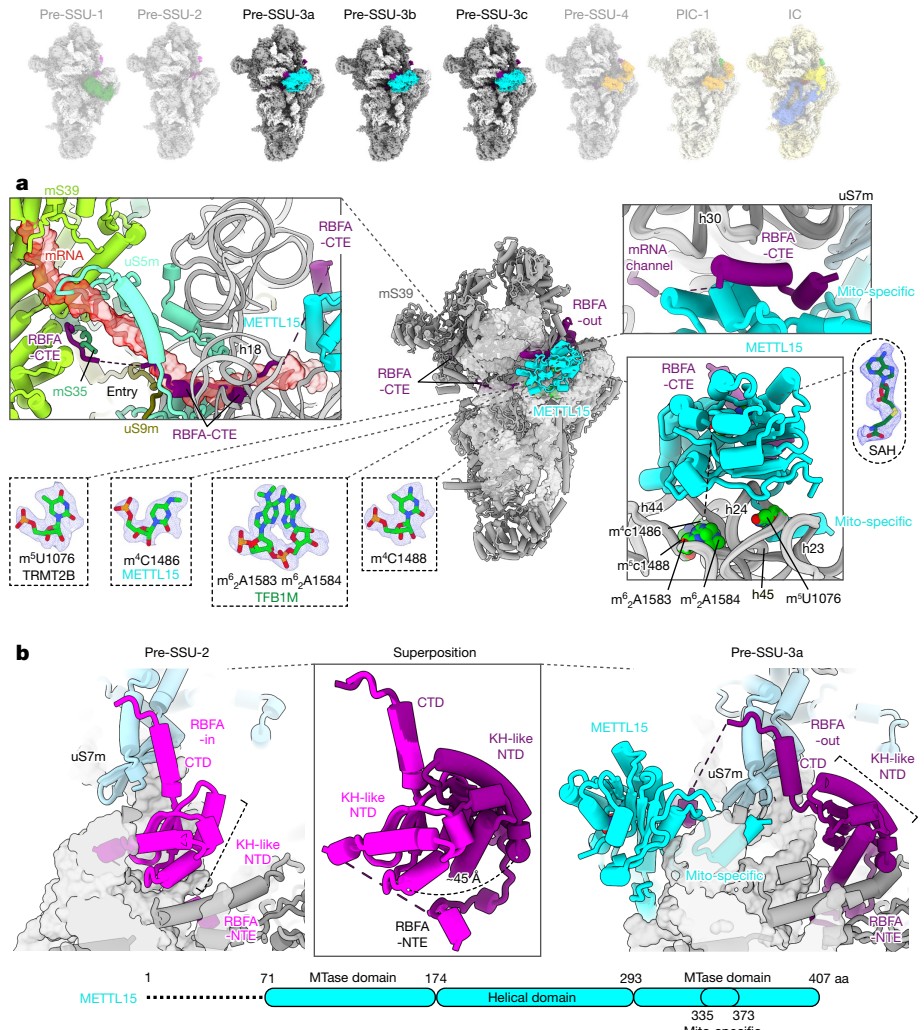

**Fig. 2 | Structures of pre-SSU-3a,b,c states. a**, Pre-SSU-3 with RBFA-out and METTL15. RBFA-CTE blocks the mRNA path on mS39, the channel entry (uS5m and uS9m) and the A site (uS5m) (upper left); mRNA is superimposed in red. RBFA-CTE interacts with METTL15 (upper right). rRNA modifications are observed in the density (bottom left). *S*-adenosylhomocysteine (SAH) with its density is away from the target residue (m⁴C1486), indicating a post-catalytic state (bottom right); the dashed line shows the 40 Å distance. **b**, Superposition of pre-SSU-2 with pre-SSU-3 shows that RBFA adopts a conformational change.

the conformational change from RBFA-in to RBFA-out. This implies that METTL15, in addition to its enzymatic activity, has an important structural role in guiding the biogenesis pathway and stabilizing the conformation that excludes the RBFA from the mitoribosomal core. The conformational change allows the head to rotate approximately 6° back from the A site towards the E site, and the associated uS7m is now placed approximately 8 Å below its initial position in pre-SSU-2 (Supplementary Video 2).

Structure prediction of RBFA by AlphaFold2 (ref. [26]) provides a conformation that is different from RBFA-in, which, to our knowledge, has been the only known conformation before this work (Extended Data Fig. 9b). We also generated the RBFA–METTL15 complex structure with AlphaFold2 by connecting the two sequences with a linker without the SSU, and the interaction through the RBFA-CTE helix was also predicted (Extended Data Fig. 9c).

To better characterize the interactions, we further classified the pre-SSU-3 particles and obtained three conformations at 2.8–3.1 Å resolution (pre-SSU-3a,b,c) that differ from each other with respect to the rRNA conformation of h23 and h24 and the METTL15–SSU interactions (Extended Data Figs. 1 and 9d and Supplementary Video 3). In pre-SSU-3a, the mito-specific insertion (residues 335–373) forms an α-helix and interacts with h23, h24 and h45 (Extended Data Fig. 9e), and

these interactions move the tip of h24 by approximately 6 Å, compared to the mature SSU, which opens the space between h24 and h44, and shifts the entire SSU platform by approximately 4 Å (Extended Data Fig. 9d,e and Supplementary Video 3). In pre-SSU-3b, the mito-specific insertion is mostly disordered, and the methyltransferase domain has looser contact to h24. As a result, h24 adopts its mature conformation, whereas h23 is extended and detached from the tip of h24 (Extended Data Fig. 9d). Finally, the pre-SSU-3c structure represents the completely folded rRNA (Extended Data Figs. 1 and 9d). Therefore, METTL15 has the additional role that facilitates rRNA folding and ultimately leads to a productive assembly pathway.

Upon completion of rRNA folding, METTL15 is released and replaced by the initiation factor mtIF3. This leads to the next assembly state pre-SSU-4 (2.9 Å resolution) (Fig. 3a, Extended Data Fig. 1 and Supplementary Video 4). Mitochondrial mtIF3 has diverged considerably from its bacterial counterpart and it was not found colocalized on the SSU with mRNA and tRNA[2]. Yet, the heart and skeletal muscle-specific loss of mtIF3 leads to cardiomyopathy, and the constitutive knockout in mice is embryonic lethal[27]. In pre-SSU-4, mtIF3 adopts a conformation with its CTD bound to rRNA h24 and h44 that blocks the premature association of LSU and fMet-tRNA^Met (Fig. 3a). The NTD contacts RBFA-CTD, whereas the association of mS37 remains sterically precluded (Fig. 3a).

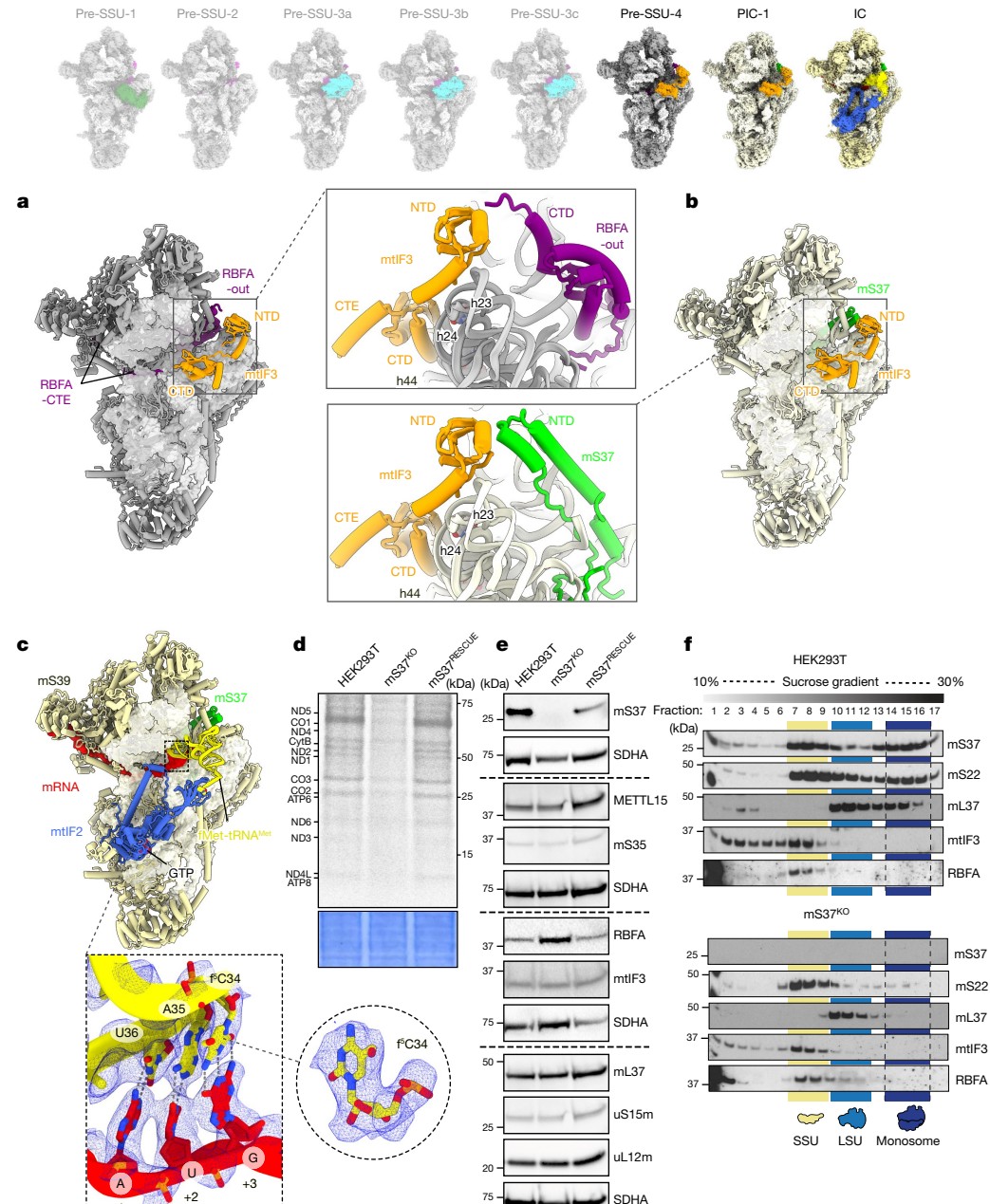

**Fig. 3 | Structures of pre-SSU-4, PIC-1 and IC states. a**, Pre-SSU-4 with RBFA-out and mtIF3. **b**, PIC-1 with mS37 and mtIF3. **c**, IC SSU–mtIF2–GTP–fMet-tRNA^Met–mRNA with the density showing codon–anticodon interactions at the P site and modification (f⁵C) on tRNA^Met. **d**, Metabolic labelling with [³⁵S]-methionine of mitochondrial translation products in the wild-type, mS37^KO and mS37^RESCUE cells. Coomassie blue-stained gel was used as the loading control. **e**, Steady-state levels of mitoribosomal proteins, assembly and initiation factors in the wild-type, mS37^KO and mS37^RESCUE cells analysed by immunoblotting. SDHA was the loading control. **f**, Mitoribosomal sedimentation on a 10–30% sucrose gradient. For **d**–**f**, representative gels from three independent biological experiments are shown. For source data, see Supplementary Figs. 4 and 5.

Consistently, knockout of the gene encoding mtIF3 shows a reduction of steady-state levels of mS37 (Supplementary Figs. 7 and 8). RBFA keeps occupying the mRNA path, and therefore the binding of mRNA remains prevented. Thus, the binding of mtIF3 during the assembly process is in stark contrast to the mechanism in bacteria, where it associates after all of the ribosomal components have been assembled to further secure the fidelity of translation initiation[28]. Recent structural studies in trypanosomes have also suggested an involvement of mitochondrial initiation factors during the assembly[29].

We also identified a class with mS37 and mtIF3 without RBFA and refined it to 2.9 Å resolution (Fig. 3b, Extended Data Figs. 1 and 3, Supplementary Fig. 2 and Supplementary Video 4). It represents pre-initiation complex (PIC-1)[2]. Therefore, the arrival of mS37 disrupts the RBFA–mtIF3 contact, and RBFA is eliminated to finalize the assembly. Unlike typical peripheral mitoribosomal proteins, mS37 is unusually conserved and can also be found in the mitoribosomal structures from trypanosomes[30], fungi[31] and ciliates[32]. To further investigate the special role of mS37, we generated its knockout cell line that showed a decrease in protein synthesis in mitochondria (Fig. 3d and Extended Data Fig. 10a). Moreover, increased steady-state levels of RBFA were observed, which supports an equilibrium shift towards assembly, as earlier states are accumulated and no monosomes were formed (Fig. 3e,f).

The structural analysis of the SSU assembly intermediates obtained from the *MRM3*-knockout model suggests that disruption of the LSU

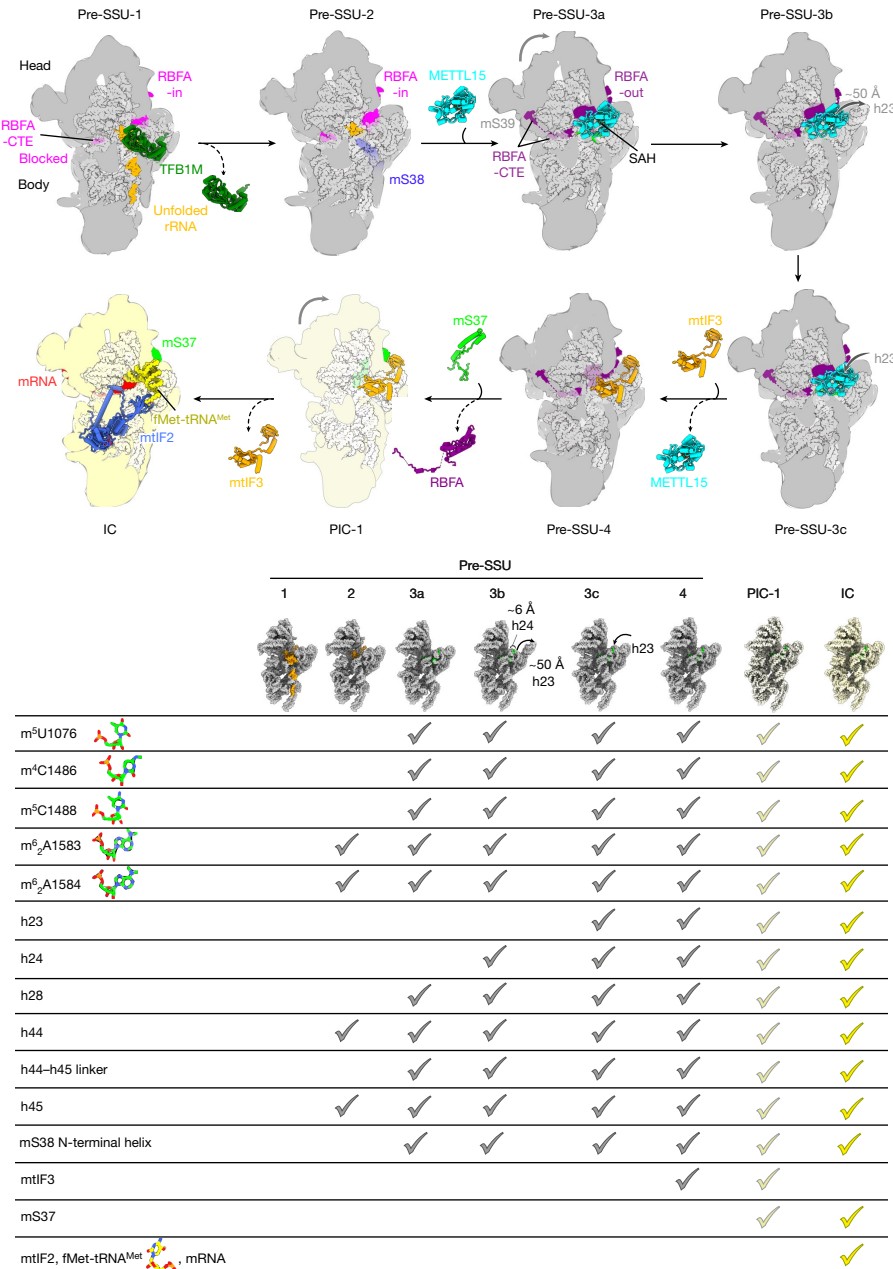

|  | Pre-SSU | | | | | | PIC-1 | IC |
|---|---|---|---|---|---|---|---|---|
|  | 1 | 2 | 3a | 3b | 3c | 4 | | |
| m⁵U1076 |  | ✓ | ✓ | ✓ | ✓ | ✓ | ✓ | ✓ |
| m⁴C1486 |  | ✓ | ✓ | ✓ | ✓ | ✓ | ✓ | ✓ |
| m⁵C1488 |  | ✓ | ✓ | ✓ | ✓ | ✓ | ✓ | ✓ |
| m⁶₂A1583 | ✓ | ✓ | ✓ | ✓ | ✓ | ✓ | ✓ | ✓ |
| m⁶₂A1584 | ✓ | ✓ | ✓ | ✓ | ✓ | ✓ | ✓ | ✓ |
| h23 |  |  |  |  | ✓ | ✓ | ✓ | ✓ |
| h24 |  |  |  | ✓ | ✓ | ✓ | ✓ | ✓ |
| h28 |  |  | ✓ | ✓ | ✓ | ✓ | ✓ | ✓ |
| h44 | ✓ | ✓ | ✓ | ✓ | ✓ | ✓ | ✓ | ✓ |
| h44–h45 linker |  |  | ✓ | ✓ | ✓ | ✓ | ✓ | ✓ |
| h45 | ✓ | ✓ | ✓ | ✓ | ✓ | ✓ | ✓ | ✓ |
| mS38 N-terminal helix |  |  | ✓ | ✓ | ✓ | ✓ | ✓ | ✓ |
| mtIF3 |  |  |  |  |  | ✓ | ✓ |  |
| mS37 |  |  |  |  |  |  | ✓ | ✓ |
| mtIF2, fMet-tRNAMet, mRNA |  |  |  |  |  |  |  | ✓ |

**Fig. 4 | Mitoribosomal SSU biogenesis.** SSU progression showing the major biogenesis states. A table of specific biogenesis events and stages when they occur is also displayed; models of rRNA are shown in insets.

assembly affects maturation of SSUs. To strengthen the structural data, we performed metabolic labelling of the nascent polypeptides in the *MRM3*-knockout cells and gradient analysis, which detected no mitochondrial translation or monosome formation (Extended Data Fig. 10a–c). We observed an accumulation of RBFA and decreased levels of mS37 in the mitoribosome (Extended Data Figs. 10c and 11a), consistent with our structural studies. Furthermore, the overall steady-state levels showed a downregulation of mS37 and increased abundance of RBFA compared to the wild type (Extended Data Fig. 11a,b). These observations were further validated by an additional analysis of cells with a knockout of the gene encoding another LSU assembly factor, GTPBP10 (refs. [33],[34]). Similarly to the *MRM3*-knockout cells, an attenuation of translation and monosome formation are correlated with decreased levels of mS37 and accumulation of RBFA (Extended Data Figs. 10a–c and 11a), supporting coordination between the assembly pathways of both subunits.

Finally, a subpopulation of 3.5% particles in our data that represents an initiation complex (IC) resolved at 3.1 Å resolution (Extended Data Fig. 1 and Supplementary Fig. 2). It has hallmark features of the initiation factor mtIF2 and initiator-tRNA that together form the complex SSU–mtIF2–GTP–fMet-tRNAMet–mRNA (Fig. 3c, Extended Data Fig. 12a and Supplementary Video 4). In addition, for the tRNAMet bound to the P site, we identified a modification at position 34 with 5-formylcytidine (f⁵C) (Fig. 3c), which is essential for mitochondrial translation[35]. The relative location of mtIF2 and the SSU head rotational state in our native complex are in a similar configuration to the reconstituted monosome IC[3], and mS37 facilitates the binding of mtIF2 (Extended Data Fig. 12b). However, the current complexes do not explain how an mRNA is delivered to the mitoribosome, thus leaving open the question of the activation mechanism of mitochondrial translation.

In conclusion, the model for the SSU progression through the assembly pathway includes a sequence of binding factors as an integrated

network, with RBFA serving as a regulator through its allosteric control mechanism (Fig. 4), which was not reported in a bacterial counterpart. The finding of the mtIF3 as an assembly factor features a new paradigm of a unique protein, mS37, that links the assembly to the initiation of translation, and additional transient intermediates are likely to exist. The role of mS37 is confirmed biochemically, and we have shown that it is reduced when the LSU is not fully assembled, supporting the special function for this protein in mitochondria.

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

# Methods

## Generation and maintenance of mammalian cell lines

The Flp-In-TRex human embryonic kidney 293T (HEK293T) cell line (Invitrogen) was cultured in Dulbecco's modified eagle medium (DMEM) supplemented with 10% (v/v) tetracycline-free FBS, 2 mM Glutamax (Gibco), 1× penicillin−streptomycin (Gibco), 50 μg ml$^{-1}$ uridine, 10 μg ml$^{-1}$ Zeocin (Invitrogen) and 100 μg ml$^{-1}$ blasticidin (Gibco) at 37 °C under 5% $CO_2$ atmosphere. Cell lines have been routinely tested for mycoplasma contamination.

For SILAC-based quantitative proteomic analysis, cells were grown in Iscove's modified Dulbecco's medium (IMDM) for Stable Isotope Labeling by Amino acids in Cell culture (SILAC) supplemented with 'heavy' ($^{15}$N-labelled and $^{13}$C-labelled) or 'light' Arg, Lys and Pro and 10% dialysed FCS (Thermo Scientific HyClone).

The *Trmt2b*-knockout cell line was cultured in RPMI 1640 medium (Gibco) supplemented with 10% FBS (Gibco), GlutaMAX (Gibco) and penicillin−streptomycin (Gibco) at 37 °C, 5% $CO_2$. Large cell culture volumes were grown in Thomson Optimum Growth 1.6-l and 5-l flasks in a shaker incubator at 110 rpm. For the cryo-EM sample, approximately 10 bln cells of the *Trmt2b*-knockout cell line were used.

## *Trmt2b* gene inactivation

For the mitoribosome preparation, we used an NS0 mouse cell line with an inactivated *Trmt2b* gene that has been characterized in ref. [16]. In brief, guide RNA (gRNA) for Cas9 targeting the first coding exon of *Trmt2b* was inserted into the pX458 vector (Addgene #48138). NS0 cells were transfected with the pX458 plasmid containing the gRNA sequence using a Lipofectamine 3000 reagent (Thermo Scientific). Twenty-four hours after transfection, GFP-positive cells were sorted using BD FACS Aria III in 96-well plates containing 0.2 ml of RPMI medium per well. Individual clones were analysed by PCR amplification of an approximately 250-bp *Trmt2b* fragment (5′-TCAAGAGTCCTAAATGCACAACC-3′ and 5′-CCAGGAGTCATCTCTACAATGC-3′) and sequencing of the amplicon. For off-target analysis, we chose five off-targets with the highest score according to the Benchling CRISPR gRNA designing tool (https://benchling.com). Each off-target was analysed by PCR amplification of approximately 250 bp and sequencing.

## *MRM3*, *mS37* and *GTPBP10* gene inactivation

To generate the *MRM3*-knockout and *mS37*-knockout cell lines, two pairs of gRNAs targeting exon 1 of *MRM3* or mS37 (Supplementary Table 4) were designed and cloned into the pSpCas9(BB)-2A-Puro (pX459) V2.0 vector to generate out-of-frame deletions. Cells were transfected with the pX459 variants using Lipofectamine 3000 according to the manufacturer's recommendations. Transfected cell populations were selected by puromycin treatment at a final concentration of 1.5 μg ml$^{-1}$ for 48 h. Subsequent to this, cells were diluted to achieve single-cell-derived clones on 96-well plates. Resultant clones were screened by Sanger sequencing to assess knockout, and loss of MRM3 and mS37 in selected clones was confirmed by western blotting. The *GTPBP10*-knockout cell line was generated as previously described[34].

## Generation of the rescue cell lines

To generate knockout cell lines stably re-expressing the protein of interest (mS37, MRM3 and GTPBP10), the HEK293T cell line that permits doxycycline-inducible expression of the gene of interest in a dose-dependent manner was used. HEK293T cells were cultured at 37 °C under 5% $CO_2$ in DMEM supplemented with 10% (v/v) tetracycline-free FBS, 1× penicillin−streptomycin (Gibco), 50 μg ml$^{-1}$ uridine, 100 μg ml$^{-1}$ zeocin (Invitrogen) and 10 μg ml$^{-1}$ blasticidin S (Gibco). Knockout cells were seeded in a 10-cm dish, 1 day before transfection, to achieve 70−90% confluency. Co-transfection of the expression plasmid pcDN5/FRT/TO (with mS37, MRM3-Flag and GTPBP10-Flag) and the Flp-recombinase plasmid pOG44 was carried out using Lipofectamine 3000 (Invitrogen), according to the manufacturer's instructions. Selective antibiotics, 100 μg ml$^{-1}$ of hygromycin (Invitrogen) and 10 μg ml$^{-1}$ blasticidin S were added 48 h post-transfection and culture media were replaced every 2−3 days. To induce expression of the protein of interest, cells were incubated with 50 ng ml$^{-1}$ doxycycline for 48 h before analysis.

## Mitoribosome preparation

Cells were harvested by centrifugation at 1,000g for 7 min. Cells were resuspended in MIB buffer (50 mM HEPES-KOH, pH 7.5, 10 mM KCl, 1.5 mM $MgCl_2$, 1 mM EDTA, 1 mM EGTA and 1 mM DTT) and allowed to swell by stirring on ice. SM4 buffer (0.28 M sucrose, 0.84 M mannitol, 50 mM HEPES-KOH, pH 7.5, 10 mM KCl, 1.5 mM $MgCl_2$, 1 mM EDTA, 1 mM EGTA and 1 mM DTT) was added to adjust the final concentrations of sucrose and mannitol to 70 mM and 0.21 M, respectively. Lysis of the cells was done in a nitrogen cavitation chamber at 500 psi for 20 min on ice. Lysate was cleared by centrifugation at 800g for 15 min. Supernatant was collected and the pellet was resuspended in half of the volume of the MIBSM buffer (70 mM sucrose, 0.21 M mannitol, 50 mM HEPES-KOH, pH 7.5, 10 mM KCl, 1.5 mM $MgCl_2$, 1 mM EDTA, 1 mM EGTA and 1 mM DTT) used at the previous step and homogenized in teflon Dounce homogenizer. Homogenate was centrifuged 800g for 15 min and supernatant was collected. The procedure with the pellet was repeated using half of the MIBSM buffer volume in the previous step. All supernatants were combined. Mitochondria were pelleted from cell lysates by centrifugation at 10,000g for 15 min at 4 °C, resuspended in MIBSM buffer and loaded on top of the step sucrose gradient (1.0 M and 1.5 M sucrose in 20 mM HEPES-KOH pH 7.5, and 1 mM EDTA) in SW40 tubes (Beckman Coulter) and centrifuged in a SW40 Ti rotor at 28,000 rpm (139,000g) for 1 h at 4 °C. Mitochondria were collected from tubes, resuspended in equal volume of 20 mM HEPES-KOH pH 7.5, and centrifuged at 10,000g for 10 min at 4 °C. Pellets of mitochondria were snap-frozen in liquid nitrogen and stored at −80 °C.

Purified mitochondria were defrosted and lysed in two volumes of lysis buffer (25 mM HEPES-KOH pH 7.5, 100 mM KCl, 20 mM Mg(OAc)$_2$, 2% Triton X-100, 2 mM DTT, 1X cOmplete EDTA-free protease inhibitor cocktail (Roche) and 40 U μl$^{-1}$ RNase inhibitor (Invitrogen) for 10 min on ice. Lysates were cleared by centrifugation at 15,871g for 10 min at 4 °C, then each 1 ml was loaded on top of a 0.4-ml sucrose cushion (0.6 M sucrose, 25 mM HEPES-KOH pH 7.5, 50 mM KCl, 10 mM Mg(OAc)$_2$, 0.5% Triton X-100 and 2 mM DTT) and centrifuged in a TLA120.2 rotor (Beckman Coulter) at 100,000 rpm (436,000g) for 1 h at 4 °C. The pellet was resuspended in resuspension buffer (25 mM HEPES-KOH pH 7.5, 50 mM KCl, 10 mM Mg(OAc)$_2$, 0.05% *n*-dodecyl β-D-maltoside (β-DDM) and 2 mM DTT), loaded on top of 10 ml of a 15−30% sucrose gradient (25 mM HEPES-KOH pH 7.5, 50 mM KCl, 10 mM Mg(OAc)$_2$, 0.05% β-DDM and 2 mM DTT) and centrifuged in a TLS-55 rotor (Beckman Coulter) at 39,000 rpm (130,000g) for 2 h 15 min. Mitoribosome gradients were fractionated into approximately 100-μl fractions and absorbance of each fraction was measured. Fractions corresponding to SSU and LSU peaks were gathered and the buffer was changed to resuspension buffer using centrifugal concentrator Vivaspin MWCO 30,000 PES (Sartoius). Obtained SSU and LSU solutions were used for grid preparation.

## SDS−PAGE and western blot analysis

To assess the steady-state levels of the individual proteins, total cell extracts or purified mitochondria were analysed using the SDS−PAGE and western blotting. Cell pellets from the wild-type HEK293T cells and *MRM3*-knockout, *mS37*-knockout and *GTPBP10*-knockout lines were lysed (50 mM Tris-HCl pH 7.4, 150 mM NaCl, 5 mM $MgCl_2$, 1 mM EDTA, 1% Triton X-100 and 1× PIC (Roche)). Protein concentration was measured using the Pierce BCA Protein Assay Kit (Thermo Fisher Scientific). Equal amounts (approximately 30 μg) of total cell extracts or mitochondrial proteins (approximately 10 μg) were separated by SDS−PAGE and subsequently wet transferred to the polyvinylidene difluoride (PVDF)

membranes for western blotting. The membranes were blocked for 1 h with 5% non-fat milk (Semper) in PBS and further incubated overnight with specific primary antibody at 4 °C. The next day, the blots were incubated with the horseradish peroxidase (HRP)-conjugated secondary antibody and visualized using enhanced chemiluminescence (ECL; Bio-Rad). The primary and secondary antibodies are summarized in Supplementary Table 4.

### [35S]-metabolic labelling of mitochondrial proteins

To label newly synthesized mitochondrial DNA-encoded proteins, cells were seeded into a six-well dish at 80–90% confluency. First, two washing steps of 5 min each in methionine/cysteine-free DMEM were performed. Subsequently, cells were incubated with fresh methionine/cysteine-free DMEM supplemented with Glutamax 100X (Gibco), sodium pyruvate 100X (Gibco), 10% dialysed FBS and 100 μg ml$^{-1}$ emetine (Sigma-Aldrich) for 20 min at 37 °C. Labelling was performed with the addition of 166,7 μCi ml$^{-1}$ of EasyTag EXPRESS [35S] protein labelling mix (methionine and cysteine) (Perkin Elmer) for 30 min at 37 °C. Following labelling, cells were washed with 1 ml of PBS three times and the final pellets were collected by centrifugation. Cells were lysed in 1× PBS-PIC with the addition of 50 units of benzonase (Life Technologies) with incubation on ice for 20 min, followed by the addition of SDS to 1% final concentration. and further incubation on ice for 30 min. After cell lysis, 30 μg total protein was separated on Bolt 12% Bis-Tris Plus (Invitrogen) SDS–PAGE gels. Gels were then incubated in Imperial Protein Stain (Thermo Fisher) for 1 h and with fixing solution (20% methanol, 7% acetic acid and 3% glycerol) for 1 h. Next, gels were vacuum-dried at 65 °C for 2 h. The resultant gel was exposed to storage phosphor screens and visualized with Typhoon FLA 7000 Phosphorimager.

### Sucrose gradient centrifugation analysis

Isolation of mitochondria and sucrose gradient centrifugation was performed as previously described[2]. In brief, 1 mg of mitochondria was lysed in lysis buffer (10 mM Tris-HCl pH 7.5, 100 mM KCl or 50 mM KCl, 20 mM MgCl$_2$, 1× PIC, 260 mM sucrose and 1% Triton X-100) freshly supplemented with 0.4 U μl$^{-1}$ final concentration of RNase Block Ribonuclease Inhibitor (Agilent), loaded onto a linear sucrose gradient (10–30%, 11 ml total volume) in 1× gradient buffer (20 mM Tris-HCl pH 7.5, 50 mM KCl, 20 mM MgCl$_2$ and 1× PIC) and centrifuged for 15 h at 79,000g at 4 °C (Beckman Coulter SW41-Ti rotor). A total of 25 fractions with a volume of 450 μl each were collected via pipetting from the top of the gradient, and 15 μl of each fraction was used for western blot analysis. For fractions 1 and 2, and for fractions 18 and 19, 7.5 μl of each fraction was combined and resolved together.

For SILAC-based proteomics, HEK293T-knockout or MRM3-kncokout cells were grown in 'heavy', containing $^{15}$N-labelled and $^{13}$C-labelled Arg and Lys, or 'light'-labelled media for more than seven doublings. Cell lines were pooled together, and mitochondrial isolation and sucrose gradient analysis were performed as described above.

### Mass spectrometry analysis

Peptides from SILAC sucrose gradient centrifugation experiments were prepared from fractions 1 and 2 joined, 3 and 4 joined, and 5 to 17 individually. Collected fractions were precipitated in 20× 100% ice-cold ethanol overnight at −20 °C. Pelleted proteins were resuspended in 6 M GuHCl/Tris pH 8.0 solution and sonicated for 5 min at maximum output (10-s on/off cycles). After a 5-min incubation at room temperature, samples underwent a second round of sonication and were later centrifuged at maximum speed for 10 min. DTT, at a final concentration of 5 mM, was added to the obtained supernatants and incubated for 30 min at 55 °C followed by incubation with 15 mM chloroacetamide for 15 min at room temperature in the dark. Before digestion, protein quantification was performed and trypsin (Pierce, trypsin protease MS-grade, Thermo Fisher Scientific) was added accordingly. Protein digestion was performed at 37 °C overnight, mildly shaking. After 12–14 h, trypsin was

inactivated using 1.2% formic acid and samples were spinned down at 3,000g for 10 min at room temperature. Samples were desalted, using desalting columns (Thermo Fisher Scientific) previously equilibrated and washed respectively with 100% acetonitrile and 0.5% formic acid, and eluted (0.5% formic acid and 50% acetonitrile). Peptides were consequently dried using SpeedVac Vacuum Concentrator and resuspended in 0.5% formic acid for mass spectrometry.

For liquid chromatography with tandem mass spectrometry analysis, peptides were separated on a 25-cm, 75-μm internal diameter PicoFrit analytical column (New Objective) packed with 1.9-μm ReproSil-Pur 120 C18-AQ media (Dr. Maisch) using an EASY-nLC 1000 (Thermo Fisher Scientific). The column was maintained at 50 °C. Buffer A and buffer B were 0.1% formic acid in water and 0.1% formic acid in 80% acetonitrile. Peptides were separated on a segmented gradient from 6% to 31% buffer B for 57 min and from 31% to 44% buffer B for 5 min at 250 nl min$^{-1}$. Eluting peptides were analysed on an Orbitrap Fusion Tribrid mass spectrometer (Thermo Fisher Scientific). Peptide precursor m/z measurements were carried out at 60,000 resolution in the 350–1,500 m/z range. The most intense precursors with charge state from 2 to 7 only were selected for higher-energy collisional dissociation (HCD) fragmentation using 27% normalized collision energy. The m/z values of the peptide fragments were measured at a resolution of 50,000 using an automatic gain control (AGC) target of $2 × 10^{-5}$ and 86-ms maximum injection time. The cycle time was set to 1 s. Upon fragmentation, precursors were put on a dynamic exclusion list for 45 s.

For protein identification and quantification, the raw data were analysed with MaxQuant version 1.6.1.0 (ref. [36]) using the integrated Andromeda search engine[37]. Peptide fragmentation spectra were searched against the canonical sequences of the human reference proteome (proteome ID UP000005640, downloaded September 2018 from UniProt). Methionine oxidation and protein N-terminal acetylation were set as variable modifications; cysteine carbamidomethylation was set as fixed modification. The digestion parameters were set to 'specific' and 'Trypsin/P'. The minimum number of peptides and razor peptides for protein identification was 1; the minimum number of unique peptides was 0. Protein identification was performed at a peptide spectrum matches and protein false discovery rate of 0.01. The 'second peptide' option was on. Successful identifications were transferred between the fractions using the 'Match between runs' option. Differential expression analysis was performed using limma, version 3.34.9 (ref. [38]) in R, version 3.4.3.

### Cryo-EM data collection and image processing

For cryo-EM analysis, 3 μl of approximately 100 nM mitoribosome was applied onto a glow-discharged (20 mA for 30 s) holey-carbon grid (Quantifoil R2/1 or R2/2, copper, mesh 300) coated with continuous carbon (of approximately 3-nm thickness) and incubated for 30 s in a controlled environment of 100% humidity at 4 °C. The grids were blotted for 3 s, followed by plunge-freezing in liquid ethane, using a Vitrobot MKIV (FEI/Thermo Fisher). The datasets were collected on a FEI Titan Krios (FEI/Thermo Fisher) transmission electron microscope operated at 300 keV with a slit width of 20 eV on a GIF quantum energy filter (Gatan). A K2 Summit detector (Gatan) was used at a pixel size of 0.81 Å or 0.83 Å (magnification of ×165,000) with a dose of 30–32 electrons per Å$^2$ fractionated over 20 frames. A defocus range of 0.8–3.8 μm was used. Detailed parameters are listed in Supplementary Tables 1 and 2.

Beam-induced motion correction was performed for all datasets using RELION 3.0 (ref. [39]). Movie stacks were motion corrected and dose weighted using MotioCor2 (ref. [40]). Motion-corrected micrographs were used for contrast transfer function (CTF) estimation with GCTF[41]. Particles were picked by Gautomatch (https://www.mrc-lmb.cam.ac.uk/kzhang) with reference-free, followed by reference-aided particle picking procedures. The reference-based pickings of SSU and LSU particles

were done separately, using their corresponding picking references. Reference-free 2D classification was carried out to sort useful particles from falsely picked objects, which were then subjected to 3D refinement, followed by a 3D classification with local-angular search. UCFS Chimera[42] was used to visualize and interpret the maps. 3D classes corresponding to unaligned or low-quality particles were removed. Well-resolved classes were pooled and subjected to 3D refinement and CTF refinement (beam-tilt, per-particle defocus and per-micrograph astigmatism) by RELION 3.1 (ref. [43]), followed by Bayesian polishing. Particles were separated into multi-optics groups based on acquisition areas and date of data collection. A second round of 3D refinement and CTF refinement (beam-tilt, trefoil and fourth-order aberrations, magnification anisotropy, per-particle defocus and per-micrograph astigmatism) were performed, followed by 3D refinement.

To classify the SSU states, non-align focus 3D classifications with particle-signal subtraction using the mask covering the factor binding were done with RELION 3.1 (Extended Data Figs. 2 and 3). The particles of each state were pooled, the subtracted signal was reverted and 3D refinement was done with the corresponding solvent mask. To improve the local resolution, the several local masks were prepared and used for local-masked 3D refinements (Extended Data Figs. 2 and 3 and Supplementary Tables 1 and 2). Nominal resolutions are based on gold standard, applying the 0.143 criterion on the Fourier shell correlation between reconstructed half-maps. Maps were subjected to B-factor sharpening and local-resolution filtering by RELION 3.1, superposed to the overall map and combined for the model refinement.

## Model building, refinement and analysis

The starting models for SSU was Protein Data Bank (PDB) ID 6RW4 (ref. [2]), whereas those of LSU were PDB IDs 6ZM5 (ref. [4]) and 5OOM[44]. These SSU and LSU models were rigid body fitted into the maps, followed by manual revision. Initial models of RBFA, TFB1M and METTL15 were generated by SWISS-MODEL[45] using PDB IDs 2E7G, 4GC9 and 1WG8, respectively, as templates. Ligands and specific extensions or insertions were built manually based on the density. Secondary-structure information prediction by PSIPRED[46] was also considered for low-resolution regions. mtIF3 and mtIF2 models were modified from previous work[2] (PDB IDs 6RW4 and 6RW5). fMet-tRNA$^{Met}$ was from PDB ID 6GAZ (ref. [3]) with the addition of the modification of f$^5$C, whereas the mRNA was manually built into the density. The CTD of bL12m was generated from PDB ID 1CTF. Coot 0.9 with Ramachandran and torsion restraints[47] was used for manual fitting and revision of the model.

Water molecules were automatically picked by Coot, followed by manual revision. Geometrical restraints of modified residues and ligands were calculated by Grade Web Server (http://grade.globalphasing.org) or obtained from the library of CCP4 7.0 (ref. [48]). Hydrogens were added to the molecules except for waters by REFMAC5 (ref. [49]).

Final models were subjected to refinement of energy minimization and atomic displacement parameters (ADP) estimation by Phenix.real_space_refine v1.18 (ref. [50]) with rotamer restraints without Ramachandran restrains, against the composed maps with B-factor sharpening and local-resolution filtering. Reference restrains were also applied for non-modified protein residues, using the input models from Coot as the reference. Metal-coordination restrains were generated by ReadySet in the PHENIX suite and used for the refinement with some modifications. Non-canonical covalent bond restrains between non-modified residues and modified residue or ligand were prepared manually and used. Refined models were validated with MolProbity[51] and EMRinger[52] in the PHENIX suite. Model refinement statistics are listed in Supplementary Tables 1 and 2. UCSF ChimeraX 0.91 (ref. [53]) was used to make the figures.

## Reporting summary

Further information on research design is available in the Nature Research Reporting Summary linked to this paper.

## Data availability

The atomic coordinates have been deposited in the RCSB PDB and the EM maps have been deposited in the Electron Microscopy Data Bank under the following accession numbers, respectively: 7PNT and EMD-13551 (pre-SSU-1), 7PNU and EMD-13552 (pre-SSU-2), 7PNV and EMD-13553 (pre-SSU-3c), 7PNW and EMD-13554 (mature SSU), 7PNX and EMD-13555 (pre-SSU-3a), 7PNY and EMD-13556 (pre-SSU-3b), 7PNZ and EMD-13557 (pre-SSU-3c), 7PO0 and EMD-13558 (pre-SSU-4), 7PO1 and EMD-13559 (PIC-1), 7PO2 and EMD-13560 (IC), 7PO3 and EMD-13561 (mature SSU), 7PO4 and EMD-13562 (pre-LSU). The following atomic coordinates were used in this study: 7BOG (*Escherichia coli* small ribosomal subunit), 6RW4 (human SSU with mitochondrial IF3), 6RW5 (human SSU with mitochondrial IF3 and IF2), 6ZM5 (human mitoribosome in complex with OXA1L), 5OOM (assembly intermediate of human LSU), 2E7G (RBFA from human mitochondrial precursor), 4GC9 (TFB1M in complex with SAM), 1WG8 (SAM-dependent methyltransferase from *Thermus thermophilus*), 6GAZ (mammalian mitochondrial translation IC) and 1CTF (L7/L12 protein). Mass spectrometry data are available via ProteomeXchange with identifier PXD031678.

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

**Acknowledgements** We thank Diamond Light Source for access to eBIC (funded by the Wellcome Trust, MRC and BBSRC) under proposal BI21643-4 with support from A. Klyszejko and Y. Song; the European Molecular Biology Laboratory for access to the cryo-EM facility under iNEXT proposal PID 4231 (funded by Horizon2020) with support from W. Hagen and F. Weis; SciLifeLab for access to the cryo-EM Swedish National Facility (funded by KAW, EPS and Kempe foundations); the Clinical Proteomics Mass Spectrometry facility of SciLifeLab; the Swedish Foundation for Strategic Research (FFL15:0325); the Ragnar Söderberg Foundation (M44/16); the European Research Council (ERC-2018-StG-805230); the Knut and Alice Wallenberg Foundation (2018.0080); the Max Planck Institute; and the Karolinska Institutet. Y.I. was supported by H2020-MSCA-IF-2017 (799399-Itohribo). J.R. is supported by the Wallenberg Fellowship Program (WAF 2017). A.A. is supported by the EMBO Young Investigator Program. P.S. is supported by Institute of Functional Genomics of the Lomonosov Moscow State University state funding.

**Author contributions** Y.I., A.K. and I.L. prepared the samples for cryo-EM. Y.I. collected and processed the cryo-EM data. Y.I. and A.K. built the models. Y.I. and A.A. carried out the structural analysis. A.K., M.C. and J.R. performed the biochemical characterization. I.A and M.C. performed the proteomic analysis. All authors contributed to data interpretation and manuscript writing.

**Funding** Open access funding provided by Stockholm University.

**Competing interests** The authors declare no competing interests.

**Additional information**

**Correspondence and requests for materials** should be addressed to Joanna Rorbach or Alexey Amunts.

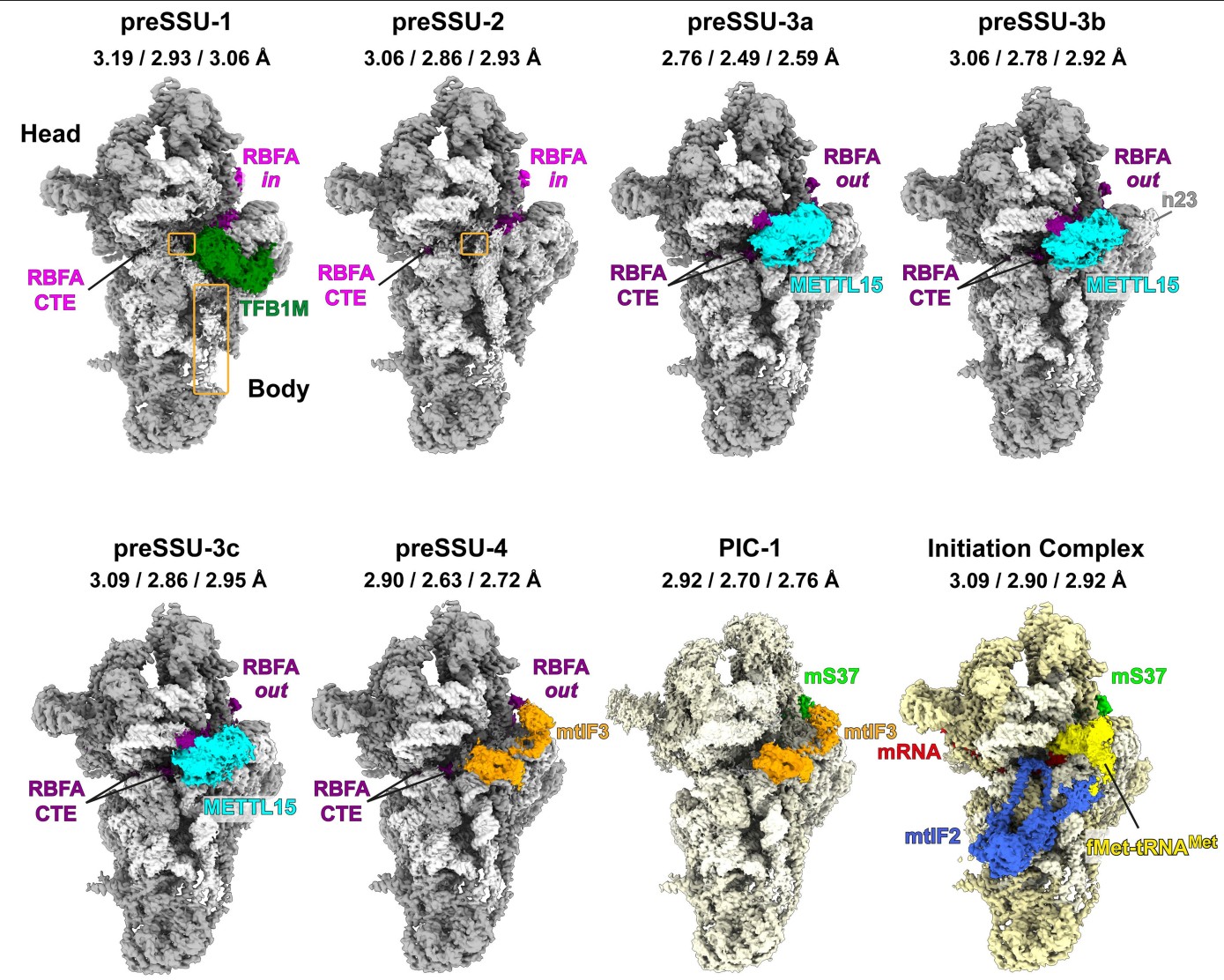

**Extended Data Fig. 1 | Cryo-EM maps of preSSU complexes with assembly factors.** The combined map of local-masked refined maps for each state with colored factors. Overall resolution and those of the local-masked refinement for shoulder and platform are listed for each state. Unfolded rRNA is indicated by orange boxes.

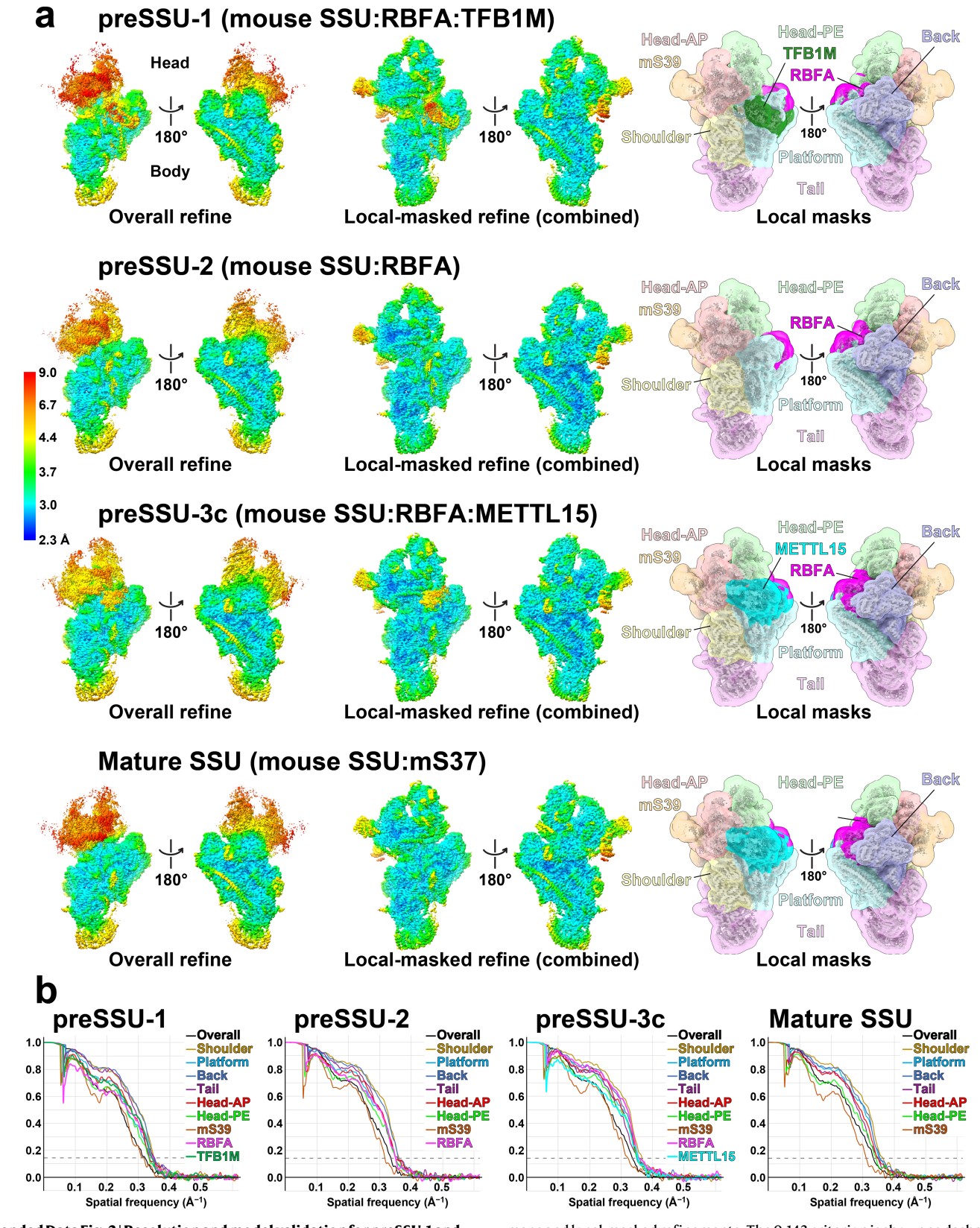

**Extended Data Fig. 2 | Resolution and model validation for preSSU-1 and preSSU-2 states. a**, Overall-refined map and combined map of the local-masked refinements colored by local resolution for each state. The local masks are shown on the right. **b**, Fourier Shell Correlation curves of the half maps and local-masked refinements. The 0.143 criterion is shown as dashed lines. The class preSSU-3c consists of particles that have the same protein content and conformation as the respective class in MRM3 knockout study presented in the Extended Data Fig. 3a.

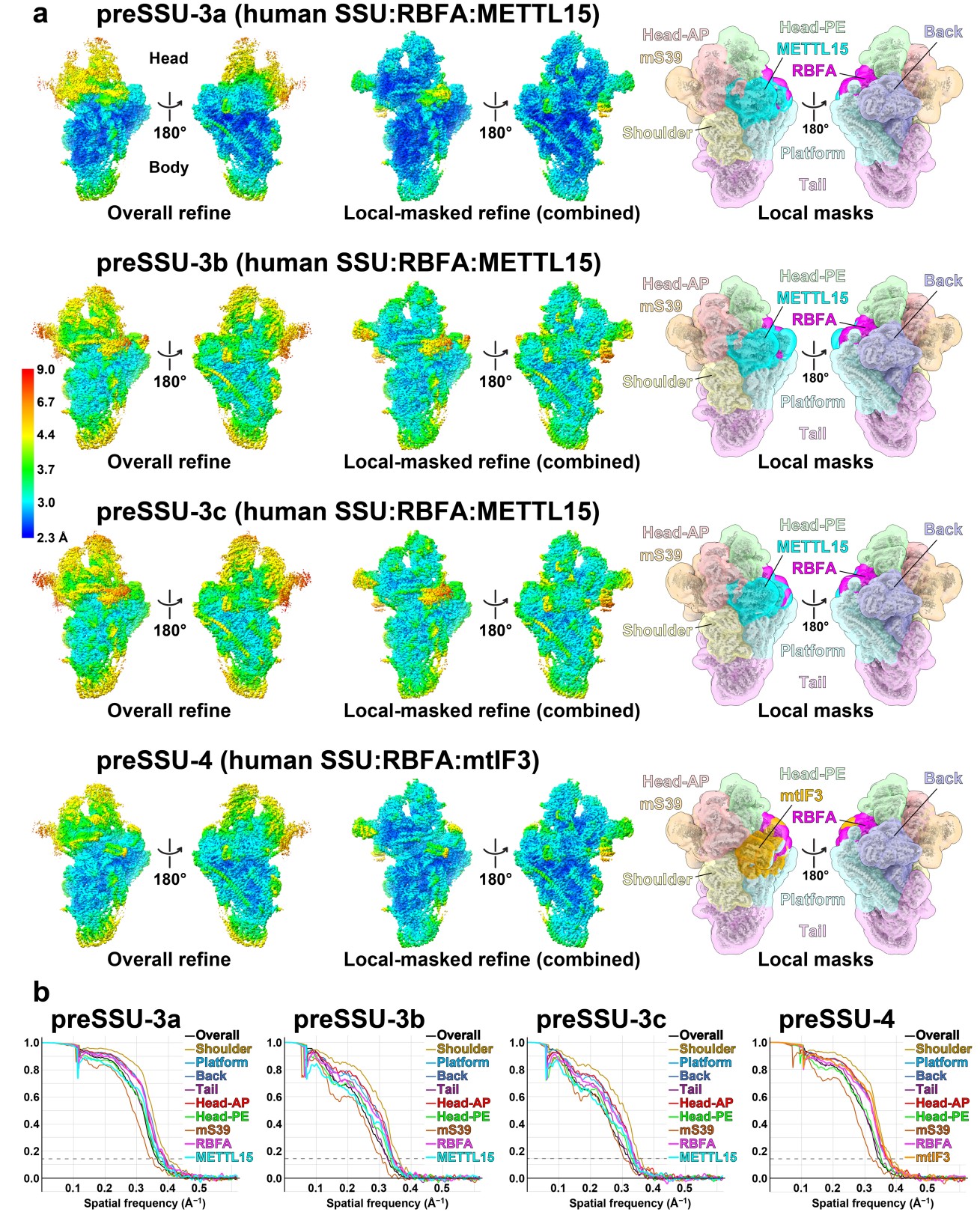

**Extended Data Fig. 3 | Resolutions and model validation for preSSU-3, preSSU-4. a**, Overall-refined map and combined map of the local-masked refinements colored by local resolution for states preSSU-3 and preSSU-4.

The local masks are shown on the right. **b**, Fourier Shell Correlation curves of the half maps and local-masked refinements for states preSSU-3 and presSSU-4. The 0.143 criterion is shown as dashed lines.

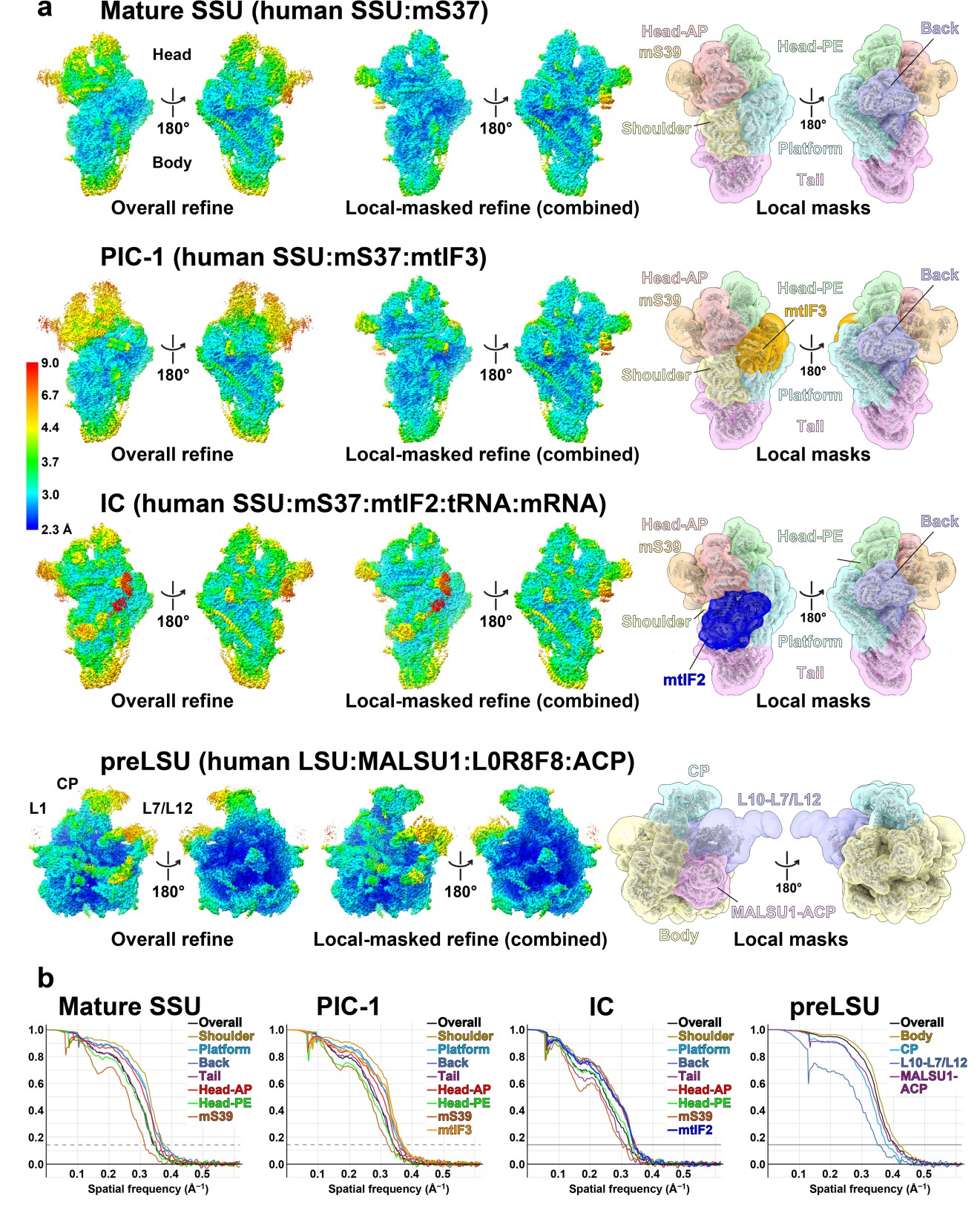

**Extended Data Fig. 4 | Resolutions and model validation for PIC-1, IC states, and LSU. a**, Overall-refined map and combined map of the local-masked refinements colored by local resolution for SSU, PIC-1, IC, LSU. The local masks are shown on the right. **b**, Fourier Shell Correlation curves of the half maps and local-masked refinements for SSU, PIC-1, IC, LSU. The 0.143 criterion is shown as dashed lines.

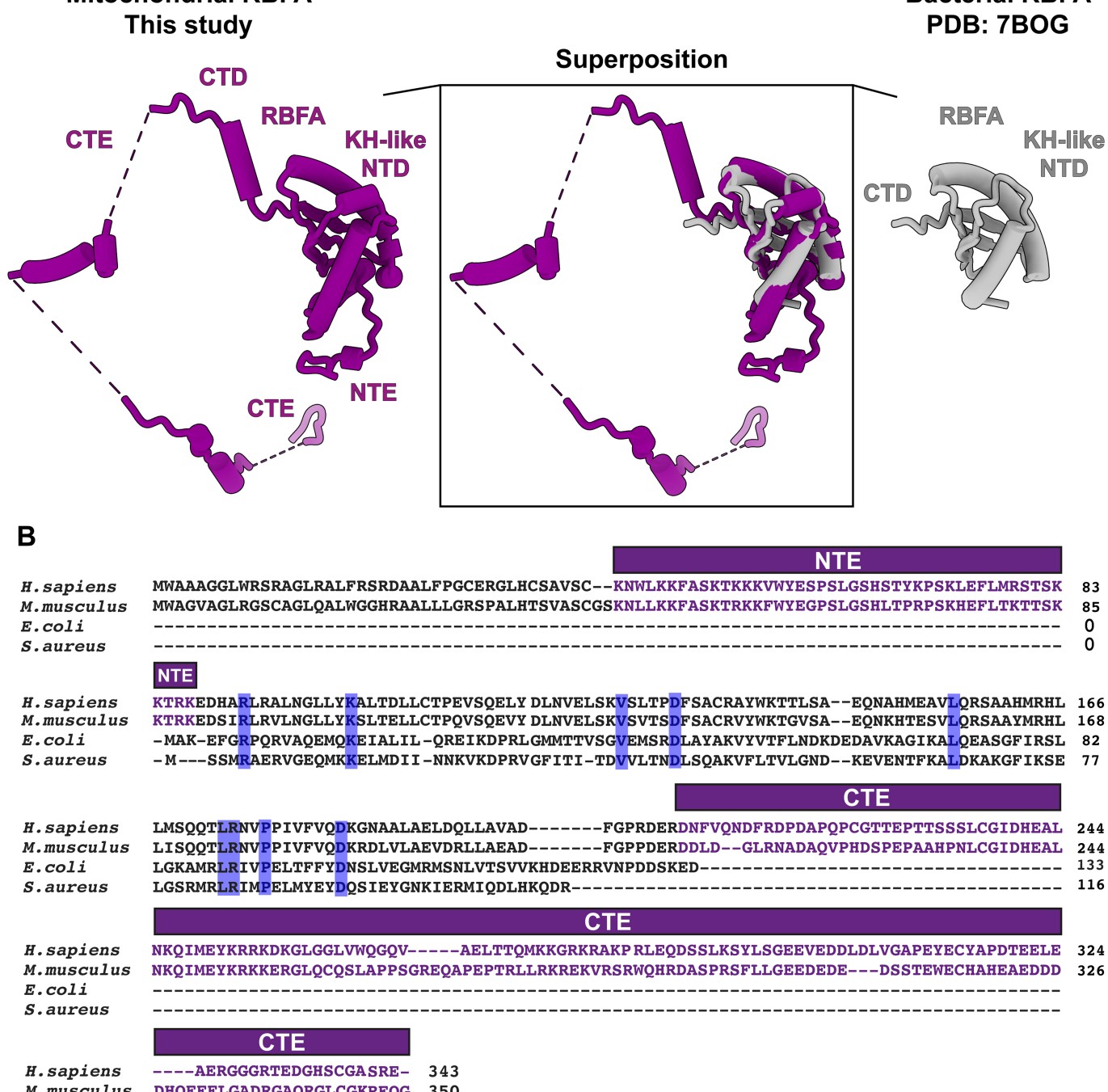

**Extended Data Fig. 5 | Comparison between mammalian mitochondrial and bacterial RBFA. a**, RBFA in mammalian mitochondria has a mito-specific N-terminal and C-terminal extensions illustrated in the structure from this study. These mito-specfic extensions are nearly twice the size of protein and contribute to multiple functions during the ribosome assembly. Inset shows the superposition of RBFA in human mitochondria and *E. coli* (PDB ID: 7BOG).

**b**, Pairwise sequence alignment of human and mouse mitochondrial RBFAs with their bacterial orthologs. The conserved amino acids are highlighted in blue. The mito-specific N and C-terminal extensions are indicated by rectangular boxes on top, and their amino acid sequences are colored in purple.

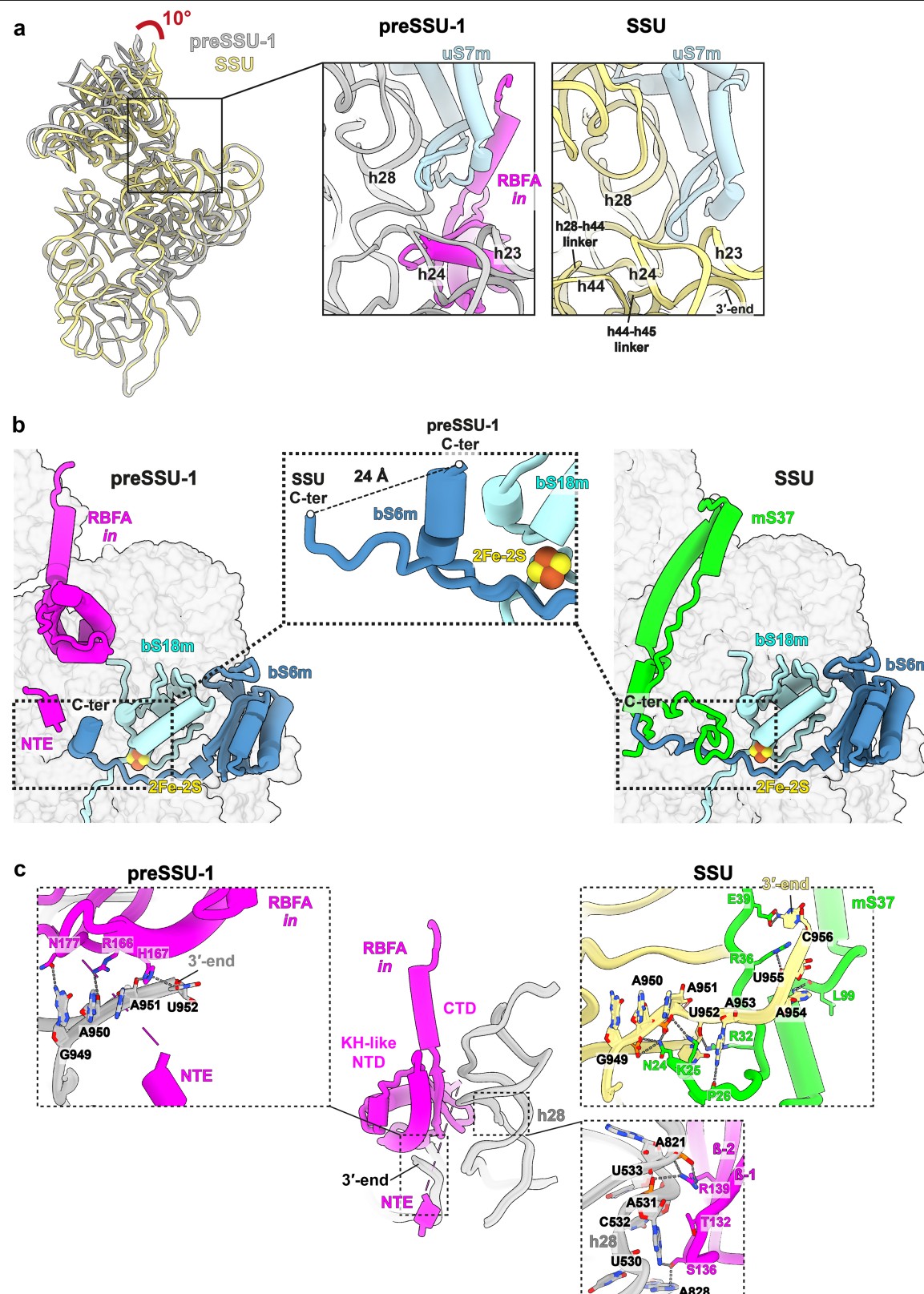

**Extended Data Fig. 6 | Structure of preSSU1 state and accompanying RBFA.**
**a**, Superposition of rRNA shows the head rotation towards the A-site that exposes the P-site region. As a result of RBFA binding, uS7m is displaced (zoom in panels). **b**, The arrangement of RBFA in preSSU1 is incompatible with mS37 in SSU. It engages the C-terminal extension of bS6m in the binding, which is dislocated by 24 Å. The formed turn is just after a mitochondria-specific iron-sulfur cluster that is coordinated by bS6m and bS18m. **c**, RBFA:rRNA contacts with SSU. Arg166, His167, and Asn177 of RBFA interact with the rRNA 3'-end, which interacts with mS37 in the mature SSU. Thr132, Ser136, and Arg139 at β1-β2 loop of RBFA interact with the rRNA h28.

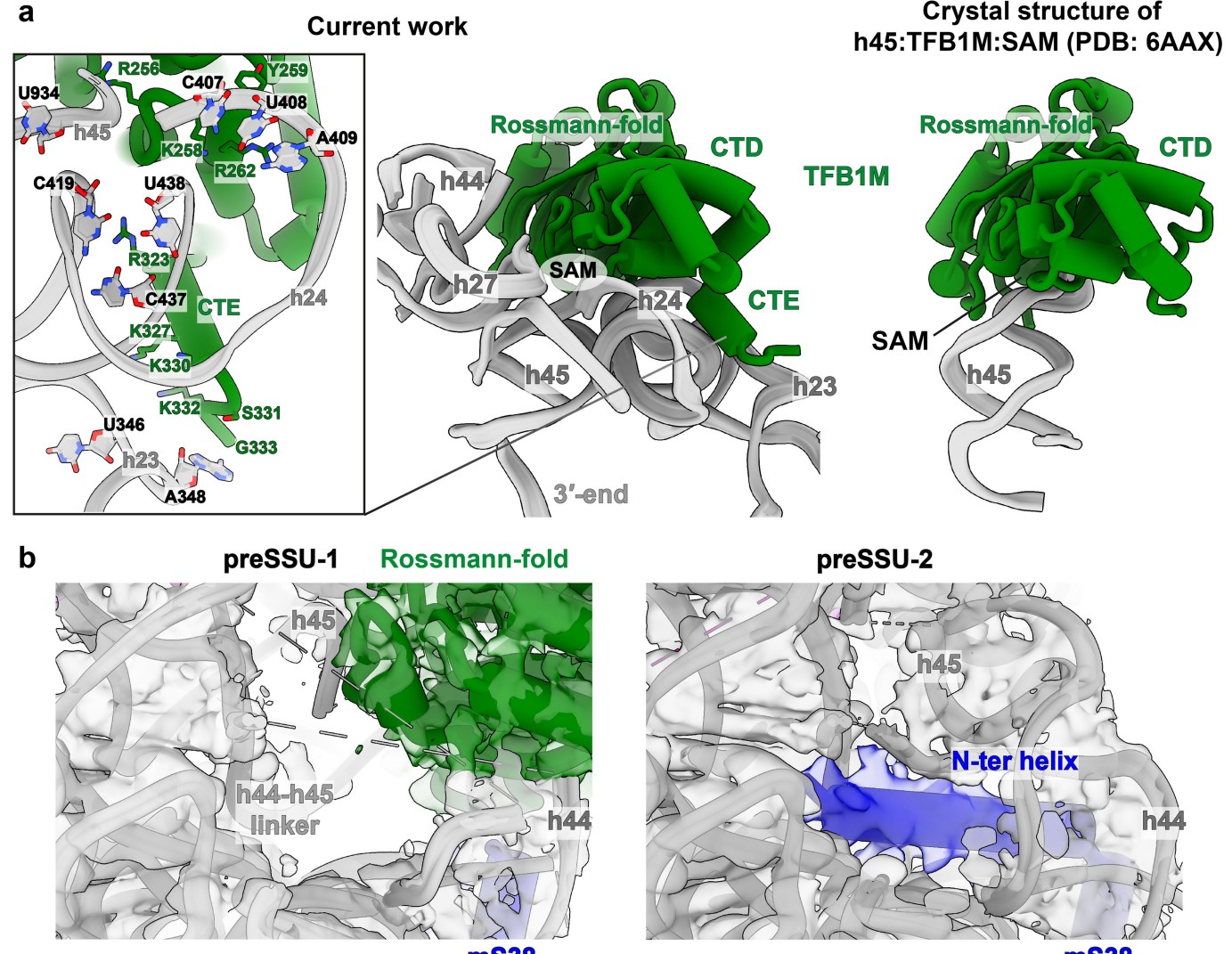

**Extended Data Fig. 7 | Structures of preSSU1 and preSSU-2 states and accompanying structural changes. a**, TFB1M:rRNA interactions and comparison with the crystal structure of h45:TFB1M:SAM[19]. TFB1M interacts with the rRNA h23, h24, h27 and h45. The C-terminal α-helix, which is disordered in the crystal structure, further links TFB1M from h24 to the platform rRNA h23 through Arg323, Lys327, Lys330 and Gly333. **b**, The cryo-EM density maps with the models, comparing the N-terminal helix of mS38 between preSSU-1 and preSSU-2. The mS38 N-terminal helix is disordered in preSSU-1, whereas it is mostly mature in preSSU-2 where rRNA is more ordered.

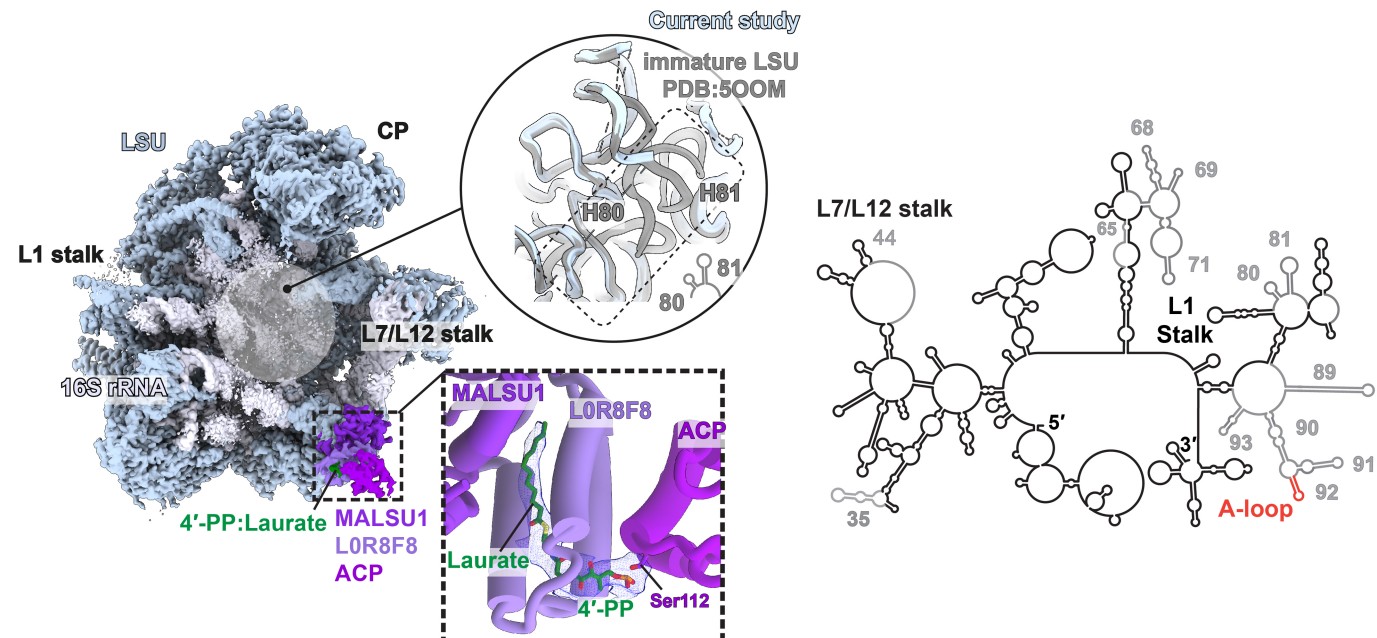

**Extended Data Fig. 8 | Cryo-EM structure of the preLSU from MRM3 knockout cells.** Cryo-EM map of preLSU with the rRNA secondary structure diagram, where the unstructured regions are gray, A-loop red. The LSU assembly intermediate is consistent with a previous native preLSU study[44], where the interfacial rRNA is unstructured, and the associated protein module MALSU1:L0R8F8:ACP prevents a premature subunit joining. In the current study, the rRNA is even more unstructured, encompassing H65-71, H89-93 with the peptidyl transferase centre, suggesting an earlier state. A density near Ser112 of the ACP that could not be accounted for by a polypeptide. The 4′-phosphopantetheine with laurate as thioester fits the density well. Therefore, ACP is found on the preLSU with its substrate. Since ACP is also required for *de novo* synthesis of fatty acid in mitochondria, as well as assembly of complex I, and iron-sulfur cluster synthesis, our study further supports that it represents a coordinative signaling molecule for metabolic sensing with regard to mitochondrial biogenesis[54,55].

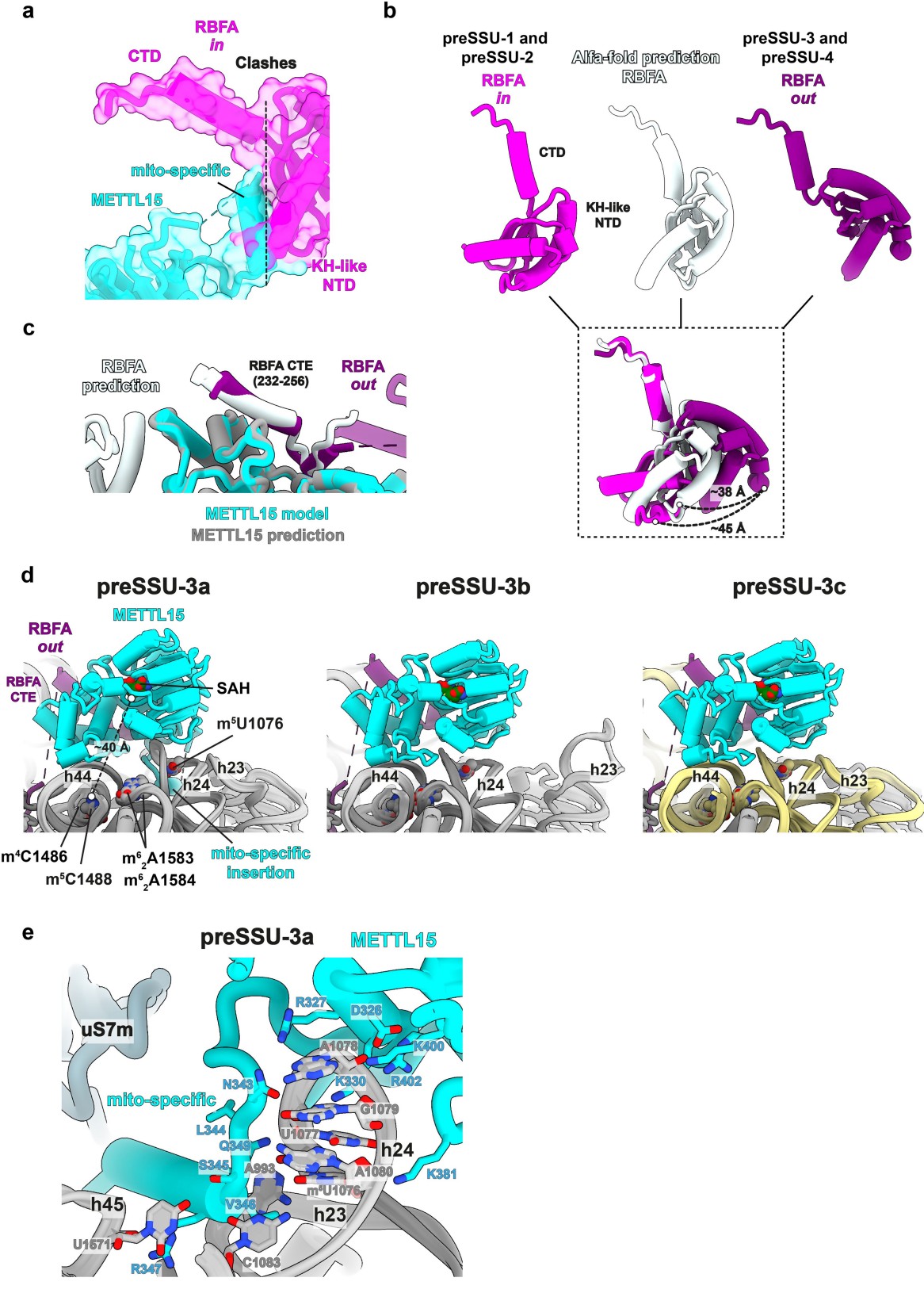

**Extended Data Fig. 9 |** See next page for caption.

**Extended Data Fig. 9 | RBFA conformational change between preSSU-2 and preSSU-3, and structures of preSSU-3a,b,c states. a**, METTL15 from preSSU3 is superimposed with RBFA-*in* from preSSU-2 showing their clash. **b**, Comparison between RBFA-*in*, RBFA-*out*, *AlphaFold2* prediction shows that the predicted conformation is different from the known structure of RBFA-*in*. **c**, RBFA:METTL15 complex prediction with *AlphaFold2* shows the interaction through the RBFA CTE helix found in the structure. **d**, In preSSU-3a, METTL15 has tight interactions with the rRNA and the mito-specific insertion forms an α-helix. The tip of h24 is shifted, which opens the space between h24 and h44, and shifts the entire SSU platform. In preSSU-3b, the mito-specific insertion is mostly disordered and METTL15 has a looser contact to h24, resulting in the mature conformation of h24, whereas h23 is extended and detached from the tip of h24. In preSSU-3c, the METTL15:rRNA interactions are the same in preSSU-2b and the rRNA is completely folded. **e**, Interactions between METTL15 and rRNA. The mito-specific insertion of METTL15 interacts with h23, h24, and h45, mainly through Asn343, Leu344 (mainchain), Val346 (mainchain) Arg347, and Asn349.

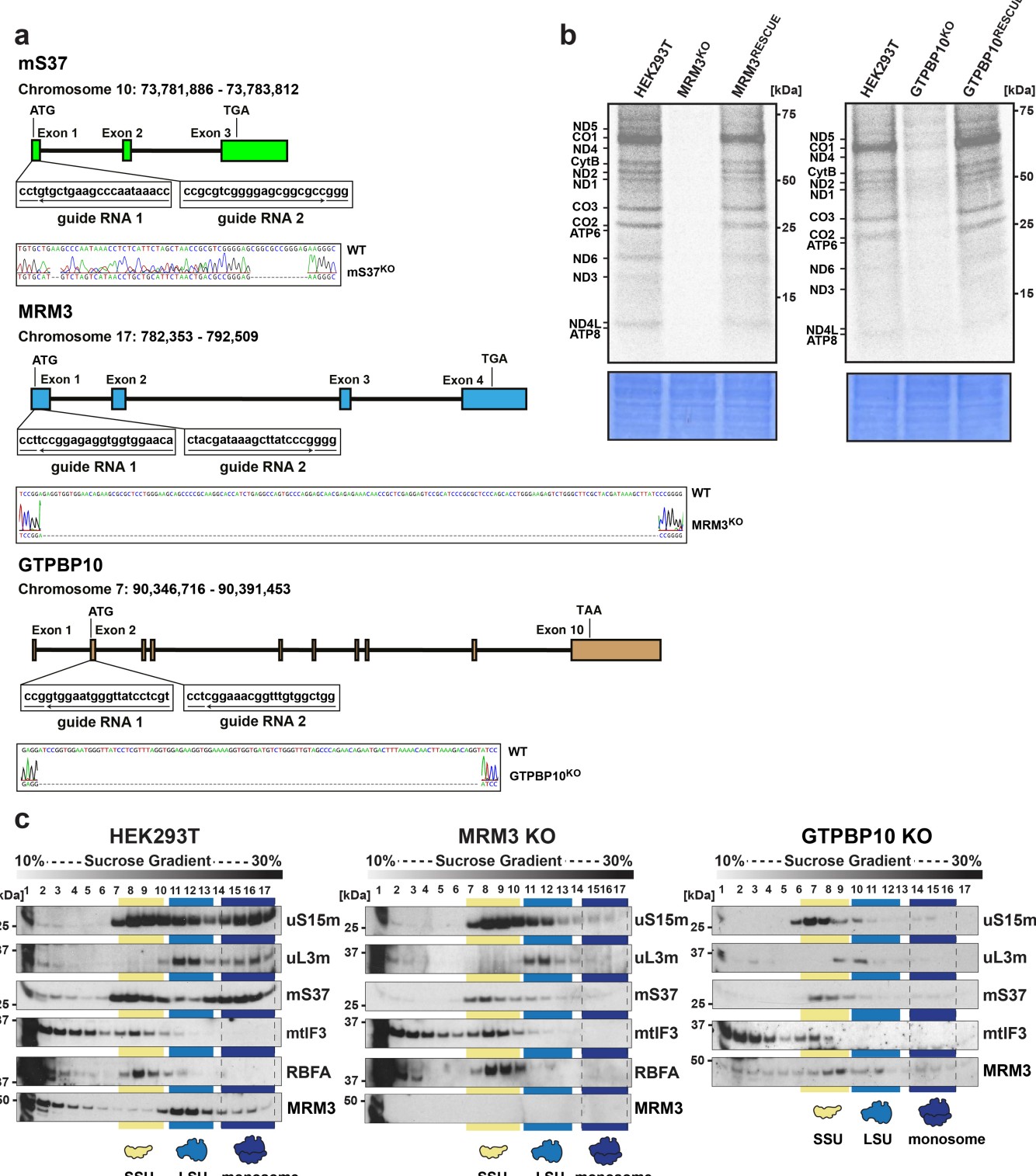

**Extended Data Fig. 10 | Biochemical characterization of MRM3 and GTPBP10 knockout cells. a**, CRISPR/Cas9 mediated targeted knockout of mS37, MRM3 and GTPBP10. The schematics illustrates the guide RNA design for targeted exon in each corresponding gene. Sequencing results and alignment with the wild type validates the knockouts indicated by bp deletions (gray dotted lines) and insertions. **b**, Metabolic labeling with [³⁵S]-methionine of mitochondrial translation products in wild type (WT) HEK293T, MRM3 and GTPBP10 knockouts and MRM3- and GTPBP10 rescue cells. Coomassie blue stained gel (bottom panel) is shown as a loading control. **c**, Mitoribosomal sedimentation on 10-30% sucrose gradient for WT HEK293T, MRM3 and GTPBP10 knockout cells. In panels b and c, representative gels from three independent biological experiments are shown. For source data, see Supplementary Fig. 6.

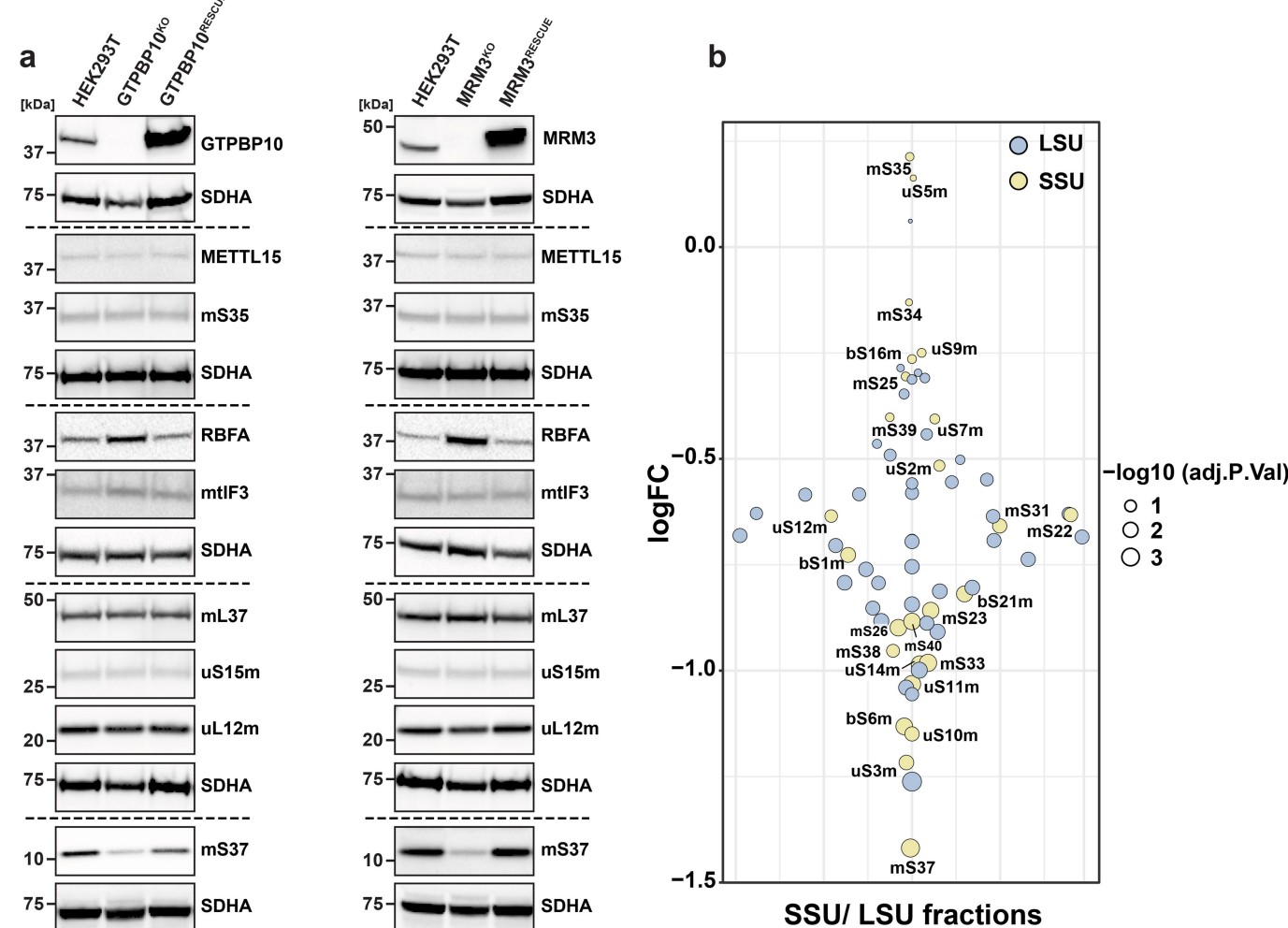

**Extended Data Fig. 11 | Steady-states levels of mitoribosomal subunits and assembly factors in GTPBP10 and MRM3 knockout cells. a**, Steady-states levels of mitoribosomal subunits and assembly factors in GTPBP10 and MRM3 knockout cells. Steady-state levels of mitoribosomal proteins (uS16m, mS35, mS37, mL37), assembly and initiation factors (MRM3, RBFA, METTL15, IF3) in the WT HEK293T, MRM3 knockout, MRM3- rescue, GTPBP10 knockout and GTPBP10-rescue cells were analyzed by immunoblotting with corresponding antibodies. SDHA was used as a loading control. Representative gels from three independent biological experiments are shown. **b**, SILAC-based proteomic analysis of sucrose gradient fractions. Protein steady-state levels of SSU (yellow) and LSU (blue) components in MRM3^KO cells are presented relative to WT HEK293T (n = 3 biologically independent experiments). Limma t-test was performed; the adjusted P-value for mS37 is 0.000521. For source data, see Supplementary Fig. 4.

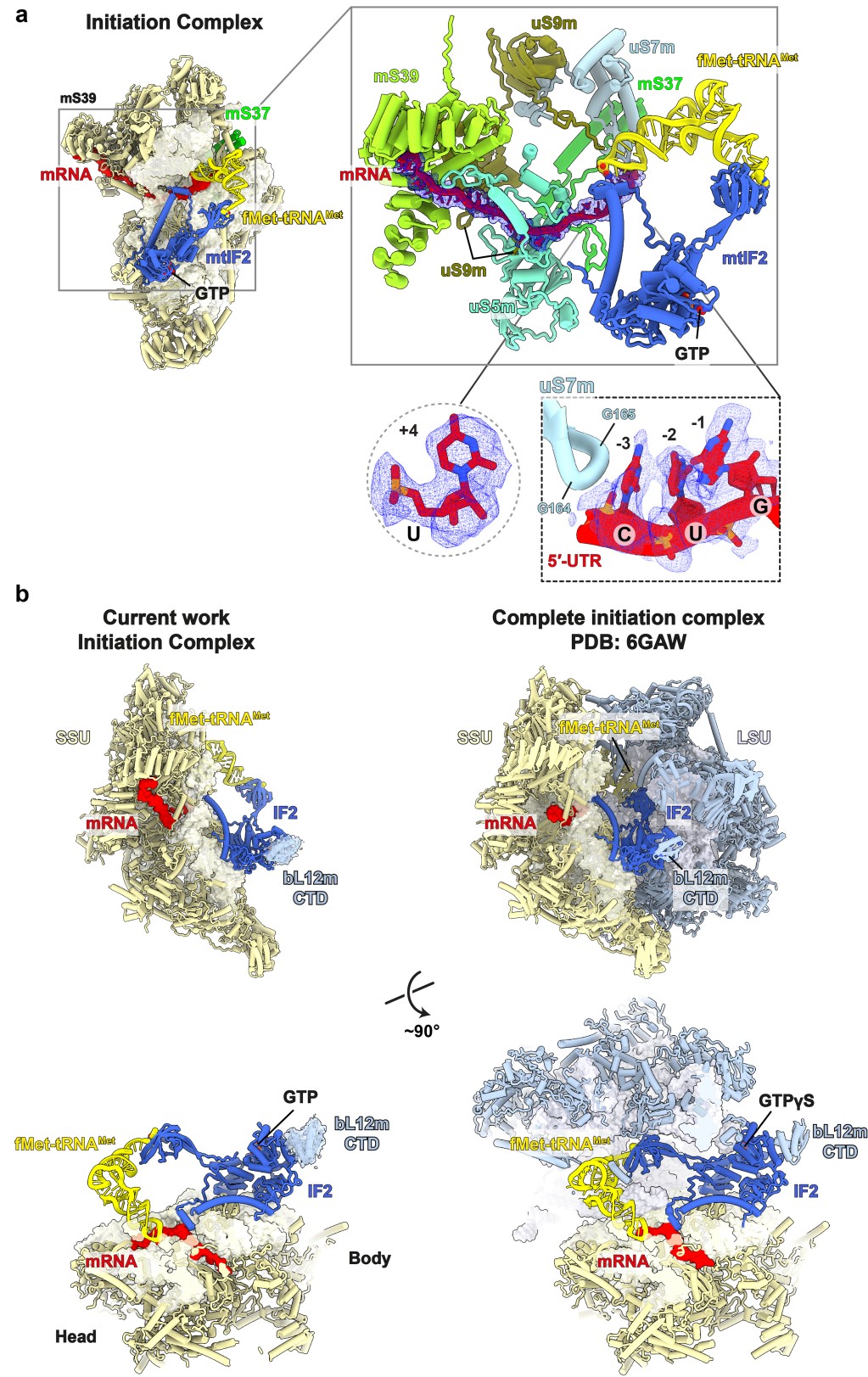

**a** Initiation Complex

mS39    mS37
mRNA
fMet-tRNA^Met
mtIF2
GTP

mS39    uS9m    uS7m    mS37    fMet-tRNA^Met
mRNA
uS9m
uS5m    mtIF2
GTP

+4
U

uS7m
G165
-3    -2    -1
G164    C    U    G
5'-UTR

**b**

**Current work**
**Initiation Complex**

SSU    fMet-tRNA^Met
mRNA    IF2
bL12m
CTD

**Complete initiation complex**
**PDB: 6GAW**

SSU    fMet-tRNA^Met    LSU
mRNA    IF2
bL12m
CTD

~90°

fMet-tRNA^Met    GTP    bL12m
CTD
IF2
mRNA    Body
Head

GTPγS    bL12m
CTD
fMet-tRNA^Met    IF2
mRNA

**Extended Data Fig. 12** | See next page for caption.

**Extended Data Fig. 12 | Structure of the Initiation Complex (IC). a**, The mRNA can be traced all the way from the mS39 docking platform, through the channel entry formed by uS5m-uS9m, A-site, P-site (pairing with tRNA^Met), and E-site (contacting uS7m). The mRNA residues at the E-site correspond to the 5′ untranslated region (UTR) that fits with a purine in the position −1, and pyrimidines in the positions −2 and −3. Out of all the mitochondrial mRNAs, only two, namely those COX1 and ND4, have the fitting residues, CUG and CCA, respectively. For the position −1, the density fits better with G than A, due to the differences in amino group locations 2 vs 6. In addition, in the start codon, the density supports AUG over AUA, and +4 fits pyrimidine, which is also present in COX1. Therefore, the cryo-EM map singles out COX1 as an enriched mRNA, associated with the resolved translation initiation complex.

Structurally, the three 5′ UTR residues in the E-site are stacked with their bases against each other and against Gly164 and Gly165 of uS7m, whereas a configuration of one or two UTR residues would not stack with uS7m in this region. The specific enrichment of the COX1 mRNA complex most likely represents the most stable variant of the pool of translation initiation complexes charged with mRNA that was trapped in our structure. **b**, Comparison with a reconstituted monosome complex. In addition to similar relative location of mtIF2 and mS37, a weak density that fits the C-terminal domain (CTD) of bL12m is observed next to the G-domain of mtIF2 (density shown as mesh). The bL12m CTD is also present in the complete initiation complex.

# Reporting Summary

## Statistics

For all statistical analyses, confirm that the following items are present in the figure legend, table legend, main text, or Methods section.

| n/a | Confirmed | |
|---|---|---|
| ☐ | ☒ | The exact sample size (*n*) for each experimental group/condition, given as a discrete number and unit of measurement |
| ☐ | ☒ | A statement on whether measurements were taken from distinct samples or whether the same sample was measured repeatedly |
| ☐ | ☒ | The statistical test(s) used AND whether they are one- or two-sided<br>*Only common tests should be described solely by name; describe more complex techniques in the Methods section.* |
| ☒ | ☐ | A description of all covariates tested |
| ☒ | ☐ | A description of any assumptions or corrections, such as tests of normality and adjustment for multiple comparisons |
| ☐ | ☒ | A full description of the statistical parameters including central tendency (e.g. means) or other basic estimates (e.g. regression coefficient) AND variation (e.g. standard deviation) or associated estimates of uncertainty (e.g. confidence intervals) |
| ☐ | ☒ | For null hypothesis testing, the test statistic (e.g. *F*, *t*, *r*) with confidence intervals, effect sizes, degrees of freedom and *P* value noted<br>*Give P values as exact values whenever suitable.* |
| ☒ | ☐ | For Bayesian analysis, information on the choice of priors and Markov chain Monte Carlo settings |
| ☒ | ☐ | For hierarchical and complex designs, identification of the appropriate level for tests and full reporting of outcomes |
| ☒ | ☐ | Estimates of effect sizes (e.g. Cohen's *d*, Pearson's *r*), indicating how they were calculated |

*Our web collection on statistics for biologists contains articles on many of the points above.*

## Software and code

Policy information about availability of computer code

| Data collection | The datasets were collected EPU 1.9 software on FEI Titan Krios (FEI/Thermofischer) transmission electron microscope operated at 300 keV with a slit width of 20 eV on a GIF quantum energy filter (Gatan). A K2 Summit detector (Gatan) was used at a pixel size of 0.81 or 0.83 Å (magnification of 165,000x) with a dose of 30-32 electrons/Å2 fractionated over 20 frames. A defocus range of 0.8 to 3.8 μm was used. A total of 5 datasets were recorded and kept. |
|---|---|
| Data analysis | Movie frames were aligned and averaged by global and local motion corrections by the program MotionCor2. Contrast transfer function (CTF) parameters were estimated by GCTF. Particles were picked by Gautomatch and 2D classified by RELION 3.0. The models were manually built with Coot 0.9 and stereochemical refinement was performed using phenix.real_space_refine in the PHENIX 1.17.1 suite. Models were predicted with AlphaFold2. The final model was validated using MolProbity. Figures were prepared with UCSF Chimera X 0.91. |

For manuscripts utilizing custom algorithms or software that are central to the research but not yet described in published literature, software must be made available to editors and reviewers. We strongly encourage code deposition in a community repository (e.g. GitHub). See the Nature Portfolio guidelines for submitting code & software for further information.

## Data

Policy information about availability of data

All manuscripts must include a data availability statement. This statement should provide the following information, where applicable:

- Accession codes, unique identifiers, or web links for publicly available datasets
- A description of any restrictions on data availability
- For clinical datasets or third party data, please ensure that the statement adheres to our policy

The atomic coordinates were deposited in the RCSB Protein Data Bank and Electron Microscopy Data bank under accession numbers: 7PNT and EMD-13551

(preSSU-1), 7PNU and EMD-13552 (preSSU-2), 7PNV and EMD-13553 (preSSU-3c), 7PNW and EMD-13554 (mature SSU), 7PNX and EMD-13555 (preSSU-3a), 7PNY and EMD-13556 (preSSU-3b), 7PNZ and EMD-13557 (preSSU-3c), 7PO0 and EMD-13558 (preSSU-4), 7PO1 and EMD-13559 (PIC-1), 7PO2 and EMD-13560 (IC), 7PO3 and EMD-135561 (mature SSU), 7PO4 and EMD-135562 (preLSU).

# Field-specific reporting

Please select the one below that is the best fit for your research. If you are not sure, read the appropriate sections before making your selection.

☒ Life sciences ☐ Behavioural & social sciences ☐ Ecological, evolutionary & environmental sciences

For a reference copy of the document with all sections, see nature.com/documents/nr-reporting-summary-flat.pdf

# Life sciences study design

All studies must disclose on these points even when the disclosure is negative.

| Sample size | 26,468 and 20,583 micrographs were analyzed for TRMT2B KO and MRM3 KO mitoribosomal structures, respectively. No statistical analyses has been performed. The number of cryo-EM particles in the single dataset collected was the number of particles available. No predetermined sample size was used for other experiments. |
|---|---|
| Data exclusions | Particles that were not mitoribosome were excluded in the analysis, since they cannot contribute to reconstruction. |
| Replication | All biochemical experiments that includes gradient purification, steady-state levels in the knock-outs and rescue cell lines were performed at least 3 times. All attempts in replication were successful. Similar cryo-EM structures were successfully obtained from preliminary datasets. |
| Randomization | Particle images were randomly assigned into half-sets to obtain gold-standard resolution estimates as described in the text. |
| Blinding | N/A to cryo-EM study; raw micrographs or particle images are not categorical data. Particles are randomly assigned into half-sets for image processing; hence no blinding is applicable. |

# Reporting for specific materials, systems and methods

We require information from authors about some types of materials, experimental systems and methods used in many studies. Here, indicate whether each material, system or method listed is relevant to your study. If you are not sure if a list item applies to your research, read the appropriate section before selecting a response.

| Materials & experimental systems | | Methods | |
|---|---|---|---|
| n/a | Involved in the study | n/a | Involved in the study |
| ☐ | ☒ Antibodies | ☒ | ☐ ChIP-seq |
| ☐ | ☒ Eukaryotic cell lines | ☒ | ☐ Flow cytometry |
| ☒ | ☐ Palaeontology and archaeology | ☒ | ☐ MRI-based neuroimaging |
| ☒ | ☐ Animals and other organisms | | |
| ☒ | ☐ Human research participants | | |
| ☒ | ☐ Clinical data | | |
| ☒ | ☐ Dual use research of concern | | |

## Antibodies

| Antibodies used | All antibodies used in this study are listed in detail in Supplementary Information Table 2. |
|---|---|
| Validation | All antibodies are commercially available and were commercially validated. |

## Eukaryotic cell lines

Policy information about cell lines

| Cell line source(s) | Flp-In TREx 293 cell line was purchased from Thermofisher Scientific.<br>NS0 murine cell line (commercial source Merck 85110503) was obtained from the research facility "Vertebrate cell culture collection" of the Institute of Cytology of the Russian Academy of Sciences, St-Petersburg. |
|---|---|
| Authentication | Flp-In™ T-REx™ Cell Line (Catalog number: R78007) was purchased from Thermofisher Scientific. No authentication is required as these cells are Zeocin and Blasticidin resistant, in contrast to any other cell line.<br>Authentification of kinetic cell morphology parameters was done for NS0 source cell line in the research facility "Vertebrate cell culture collection" of the Institute of Cytology of the Russian Academy of Sciences, St-Petersburg. |

| Mycoplasma contamination | Cell line tested negative for mycoplasma contamination. |
| --- | --- |
| Commonly misidentified lines (See ICLAC register) | No commonly misidentified cell lines were used in the study. |

