## [Peer Review File · Nature]

Manuscript Title: Mechanism of mitoribosomal small subunit biogenesis and preinitiation

Reviewer Comments & Author Rebuttals

Reviewer Reports on the Initial Version:

Referee #1:

The manuscript by Itoh et al. describes the results of a structural investigation of the human small mitoribosomal subunit (SSU) late-stage assembly and its articulation with the translation initiation process. The study reports a wealth of new structures that recapitulate the step-wise maturation of the SSU at a late stage, involving several auxiliary factors that assemble sequentially and culminate in the binding of IF3, a hallmark of the commencement of the initiation process. The study also reveals how a ribosomal protein (mS37) adapted to link the maturation with the initiation process through the binding interplay with IF3.

One technical aspect that merits to be highlighted is the depletion of an rRNA methyltransferase (TRMT2B), which enabled the accumulation of several pre-SSU complexes in different states, including assembly factors and lacking several ribosomal proteins. This is the first structural study to show SSU assembly intermediates for the human/mammalian mitoribosome.

The analysis made of the obtained structures is insightful and reveals numerous novel aspects mainly related to the roles and functions of several maturation factors. The manuscript is well written and the figures are for the most part clear and even aesthetic. The methods are sufficiently detailed and the Extended Data are all justified. References are appropriate.

I recommend its publication with minor corrections.

For such a beautiful manuscript, very easy to review, I only have few minor points, mainly related to the figures:

Fi. 1: In panel a, the RBFA CTE atomic model is shown inside a transparent magenta density. The density needs to be made more transparent, as the stomic model inside is hardly visible. mS37 is shown in transparent green in order to highlight the clash with RBFA; however, the display is a bit awkward... Perhaps mS37 needs to be made more transparent to drive the point more clearly, and the authors may want to put some collision indications (colliding arches, etc...). For the whole figure, many labels overlap with the structure; when possible, the authors may want to avoid large labels that cover up parts of the structure.

Fig. 2: Transparent densities with outlined contours are shown for several modified nucleotides. Although one could appreciate the resolution of the map through this display, it is at times difficult because of the high level of transparency of the densities. I suggest increasing the opacity a bit to

give them some color in order to better appreciate their features. Also, avoid labels that cover up parts of the structures.

Fig. 3: Panel c shows transparent density with outlined contours for several nucleotides. Similarly to the point above, I suggest increasing the opacity a bit to give them some color in order to better appreciate their features.

In their conclusion, the authors state the following: “The finding of the repurposed IF3 as an assembly factor features a new paradigm of a unique protein mS37, that links the assembly to the initiation of translation.” Based on the presented data, the conclusion is sound. However, do the authors exclude the existence of a short-lived transient PIC, between the PIC-1 and the IC, where IF3 can be bound alongside IF2, tRNA^{fMet} and the mRNA? In other words, wouldn't it be reasonable to imagine that rather than being “repurposed”, IF3 has acquired a second role? That is, IF3 dissociates and returns with IF2 in the context of the PIC... Could the authors comment?

Yaser Hashem

Referee #2:

Itoh et al. show several structural snapshots of assembly intermediates of the mitochondrial small ribosomal subunit. This enables the authors to provide an assembly scheme with the stepwise progression of SSU maturation and the molecular function of involved factors. To achieve this, Itoh and colleagues isolated ribosome complexes/intermediates from assembly factor-deficient cells (murine TRMT2B KO; human MRM3 KO) and solved structures by cryo electron-microscopy. They solved eight different states with three associated assembly factors (RBFA, TFB1M, METTL15) and the two initiation factors, mtIF2 and mtIF3. They were able to place mS37 within the assembly pathway and show that it is the last MRP which joins the SSU complex during maturation. The authors show some exciting findings, e.g. 1. mtIF3 acts during assembly, which is in contrast to its bacterial homolog and the study of Rudler et al. (2019); 2. RBFA functions as a regulator occupying the mRNA channel during maturation; and 3. they solved an initiation complex bound to COX1 mRNA isolated from MRM3 KO. The provided data are of very good quality and certainly of high interest in the field of mitochondrial ribosome biogenesis and translation.

However, as it is written right now I would have some doubts that it is appealing for other scientists, as sometimes the transition is not comprehensible and requires a lot of expertise to follow. For example, it is not clear why the authors have used the MRM3 KO (cell line deficient in a methyltransferase of the LSU) to study the biogenesis of the SSU. It might be difficult to follow this logic, especially if you are not from the field. The same is true for including the GTPBP10 KO. The associated results of GTPBP10 KO are not explained and non-experts will be lost as there is no background information. The authors also describe an assembly intermediate of the LSU isolated from MRM3 KO. This comment is somehow misplaced and disturbs the flow of the text (line 185-189). Thus, I would recommend some changes in the text to improve the flow and the understanding for a broad readership.

Apart from that I have some comments and questions, which the authors might want to address:

1.) I was wondering whether the authors consider possible off-target pathways when using assembly factor-deficient cell lines. This is often a big concern in the field of bacterial ribosome assembly that intermediates are accumulating under certain KO conditions which are not able to mature further. Can the authors exclude this possibility? It might be worth to include a short statement/little discussion in the manuscript.

2.) The authors have used several knockout cell lines. As they have sequenced them, it would be good to provide the mutations resulting in the loss of the protein. In my opinion it is always good to provide a corresponding rescue cell line to prove the specificity of the phenotype associated with the knockout.

3.) It is not clear how the authors have generated the GTPBP10 KO cell line. Page 12/line 265 “as previously described”: there is no associated reference. In the main text they refer to Cheng et al. (2021); however, in this study a GTPBP10 KO was not generated, only a FLAG-tagged version of GTPBP10 was described. Thus, the authors should clarify this.

4.) I was wondering why they could solve specifically the SSU with bound COX1 mRNA from MRM3 KO. Does it mean that COX1 mRNA is more abundant/stable than others in MRM3 KO? Why would this be the case? Any explanation or speculation?

Minor points:

Lines 40-41: messenger (mRNA) and transfer (tRNA) > messenger RNA (mRNA) and transfer RNA (tRNA)

Line 51: ribosomal (rRNA) > ribosomal RNA (rRNA)

Lines 222-223, “ mS37...is lacking when the LSU is not fully assembled,...”: I think this is not accurate as there is still mS37 detectable in MRM3 KO and GTPBP10 KO gradients (Ext. Data Fig. 13b). Better to use “reduced” or similar terms.

Referee #3:

In this interesting and high-quality study, the authors examine the biogenesis of the small mitoribosome subunit using cryo-EM. By determining the structures of a series of intermediates assembly conformations, they reveal the role of the sequential recruitments of assembly factors and the key role played by mS37 to establish a translation initiation competent subunit. The study reveals: i) a crucial role of RBFA that undergoes a series of structural changes allowing the sequential recruitment of other assembly factors, while blocking interaction of mS37; ii) interaction of IF3 preceding the last step of assembly; iii) the paramount role of mS37 as the component that links assembly to initiation of translation. The novelty of this work is that it sheds light on the mechanistic role of RBFA and mS37 and structural defines intermediate step of the assembly of the small subunit

of the mitoribosome. This contributes to a refined understanding of the mechanisms underlying mitoribosome assembly, and of differences compared to the bacterial counterpart.

Specific comments

1) The intermediates are captured by depleting TRMT2B or MRM3. Can the authors comment on how well these strategies block assembly in physiological states? In other words, how do the intermediates identified here depend on the depletion of specific components?

2) The authors identify IF3 as an assembly factor (it would be better to name it mtIF3). What happens to mitoribosome assembly if IF3 is deleted? Is IF3 really an assembly factor or rather together with mS37 characterizes an earlier status of the initiation complex? A previous paper from the authors (PMID: 32522994) identifies already the interaction between mS37 and mtIF3 as an initial pre-initiation complex. What is the novelty here? Are the authors proposing now a different model?

3) Fig. 3d-f: How many times have these experiments been repeated?

Author Rebuttals to Initial Comments:

We thank the Referees for taking the time to provide constructive suggestions on how to improve the study and its presentation. We addressed all the requests and followed the valuable suggestions. In the revised version, the following changes have been made:

- 1) rescue experiments for all the knockout cell lines have been performed, and the results are added, as requested by Referee #2;
- 2) mtIF knockout investigation has been performed, as requested by Referee #3;
- 3) all the main figures are improved according to the guideline provided by Referee #1;
- 4) limitations of the study are spelled out, roles of some of the factors are described more carefully, and the flow between the sections was improved following the guidelines provided by all the Referees.
- 5) four supplementary movies are added to further drive the discussed points more clearly.

Below is the point-by-point response.

Referee #1:

For such a beautiful manuscript, very easy to review, I only have few minor points, mainly related to the figures:

Fig. 1: In panel a, the RBFA CTE atomic model is shown inside a transparent magenta density. The density needs to be made more transparent, as the stomic model inside is hardly visible. mS37 is shown in transparent green in order to highlight the clash with RBFA; however, the display is a bit awkward... Perhaps mS37 needs to be made more transparent to drive the point more clearly, and the authors may want to put some collision indications (colliding arches, etc...). For the whole figure, many labels overlap with the structure; when possible, the authors may want to avoid large labels that cover up parts of the structure.

We followed the detailed guideline provided by the Referee to improve Fig. 1, and the following requested changes have been implemented:

- The surface is made more transparent to make the RBFA CTE atomic structure more visible. The surface here is generated from the model, and we now indicated it in the legend.
- mS37 is made more transparent.
- We tried adding colliding arches to indicate the clash, but it appears to be confusing with the structural details. Therefore, to comply with the Referee's comment, we produced a Supplementary Movie 2 to drive the point more clearly.
- The following labels are adjusted not to overlap with the structure: RBFA-in, TFB1M, uS7m, bS6m, bS1m, uS11mm, KH-like NTD, Unfolded RNA, mS38.

Fig. 2: Transparent densities with outlined contours are shown for several modified nucleotides. Although one could appreciate the resolution of the map through this display, it is at times difficult because of the high level of transparency of the densities. I suggest increasing the opacity a bit to give them some color in order to better appreciate their features. Also, avoid labels that cover up parts of the structures.

We followed the suggestion and increased the opacity of the densities, as well as adjusted the following labels to avoid the cover up of structures: panel A) uS5m, mS39, METTL15, uS7m, RBFA-CTE, mito-specific; panel B) uS7m, CTD; we also deleted the labels "entry", repeating RBFA-CTE, METTL15, 40 Å.

To further clarify the point, we also produced Supplementary Movie 3 that features each one of the mentioned densities.

Fig. 3: Panel c shows transparent density with outlined contours for several nucleotides. Similarly to the point above, I suggest increasing the opacity a bit to give them some color in order to better appreciate their features.

As suggested, the opacity of the density has been increased by a factor of 2.5, and colors have been adjusted. In addition, in Fig. 4 we removed repetitive labels “exit”, “entry”, and several other labels to clarify the images. Thank you for the suggestions to improve the figures.

In their conclusion, the authors state the following: “The finding of the repurposed IF3 as an assembly factor features a new paradigm of a unique protein mS37, that links the assembly to the initiation of translation.” Based on the presented data, the conclusion is sound. However, do the authors exclude the existence of a short-lived transient PIC, between the PIC-1 and the IC, where IF3 can be bound alongside IF2, tRNA^{fMet} and the mRNA? In other words, wouldn't it be reasonable to imagine that rather than being “repurposed”, IF3 has acquired a second role? That is, IF3 dissociates and returns with IF2 in the context of the PIC... Could the authors comment?

Indeed, cryo-EM captures the most stable states, and thus the existence of short-lived transient complexes that escaped detection cannot be excluded, even if the current experimental data doesn't provide evidence. Therefore, to comply with the Referee's comment, we: 1) removed “repurposed” from page 10, line 224; 2) added a concluding remark on page 10, lines 225-226 “additional transient intermediates are likely to exist”; 3) added “stable” on page 2, line 54.

Referee #2:

However, as it is written right now I would have some doubts that it is appealing for other scientists, as sometimes the transition is not comprehensible and requires a lot of expertise to follow. For example, it is not clear why the authors have used the MRM3 KO (cell line deficient in a methyltransferase of the LSU) to study the biogenesis of the SSU. It might be difficult to follow this logic, especially if you are not from the field.

We thank for the suggestions to improve the organisation of the paper in order to make it easier to follow. We implemented the suggested change by adding a paragraph that introduces the rationale for using the MRM3 KO cell line on pages 4-5, lines 106-111:

“In the same dataset, we also observed a class with a subset of particles with folded rRNA and the additional density that is different from TFB1M, suggesting a later sequential assembly intermediate. To explore how the SSU progresses through the mitoribosomal assembly pathway after TFB1M maturation, we reasoned that an inhibition of the corresponding LSU assembly would allow the SSU to proceed towards later stages up to the subunit joining. Therefore, next we knocked out the enzyme mitochondrial rRNA methyltransferase 3 (MRM3) responsible for a late assembly of the LSU (21), and collected cryo-EM data of the accumulated SSU and LSU particles (Extended Data Figs. 4-6, Extended Data Fig. 10).”

In addition, we composed Supplementary Movie 1 that provides the context for all the structural data obtained in this study and put the MRM3 KO related structures in the sequence.

The same is true for including the GTPBP10 KO. The associated results of GTPBP10 KO are not explained and non-experts will be lost as there is no background information.

We followed the suggestion, and a paragraph explaining the associated results of GTPBP10 KO has been added with references to the recent studies on page 9, lines 206-211:

“These observations were further validated by an additional analysis of cells with a knockout of another LSU assembly factor, GTPBP10 (33-35). Similarly to the MRM3 knockout, an attenuation of translation and monosome formation are correlated with decreased mS37 levels and accumulation of RBFA (Extended Data Figs. 13, A to C and 14A), supporting coordination between the assembly pathways of both subunits.”

The authors also describe an assembly intermediate of the LSU isolated from MRM3 KO. This comment is somehow misplaced and disturbs the flow of the text (line 185-189). Thus, I would recommend some changes in the text to improve the flow and the understanding for a broad readership.

We followed the recommendation to improve the flow of the text and moved the mentioned comment to the legend of Extended Data Fig. 10, where the LSU structural data is presented.

The quality of the map for the LSU part further allowed us to identify a density for 4'-phosphopantetheine with laurate substrate on the ACP that could not be resolved in any of the previous studies.

Apart from that I have some comments and questions, which the authors might want to address:

1.) I was wondering whether the authors consider possible off-target pathways when using assembly factor-deficient cell lines. This is often a big concern in the field of bacterial ribosome assembly that intermediates are accumulating under certain KO conditions which are not able to mature further. Can the authors exclude this possibility? It might be worth to include a short statement/little discussion in the manuscript.

We thank the Reviewer for raising this important concern that allows us to expand on the description of the experimental design used in the study, as well as to add a comment on the limitation of the study. We agree that the present and any cryo-EM study aiming to characterise intermediates under KO conditions might be biased towards stable complexes, and thus the overall picture might not necessarily represent a complete native productive path. Since the possibility of the off-target pathways cannot be excluded, we added in a concluding sentence on page 10, lines 225-226 *“additional transient intermediates are likely to exist”*.

To reduce the risk of possible off-target pathways, our experimental setup was based on two knockout systems of two different subunits: TRMT2B KO for SSU, and MRM3 KO for LSU. Upon particle classification, at least one identical assembly class, namely preSSU-3c was detected in each one of the two independent experimental systems (Extended Data Fig. 3A and 6A). In addition, a competent initiation complex was detected as one of the products of the pathway. While not completely removing the risk of an off-target pathway, these data support a productive path. We now introduced this information for a reader, where the MRM3 KO data is presented, on page 5, lines 111-115: *“~7% of the SSU particles in this cryo-EM data showed identical structural features and presence of the same additional density that were also detected in the particles from the TRMT2B-depleted cells, and the latter was identified as the N4-methylcytidine methyltransferase METTL15 (Extended Data Fig. 1, 3A, 6A)”*.

In addition, a comment was also added to the legend of Extended Data Fig. 3 “*The class preSSU-3c consists of particles that have the same protein content and conformation as the respective class in MRM3 knockout study presented in the Extended Data Fig. 6A.*”

Since a colocalization of assembly and initiation factors (although different than in mammalian system) has also been reported in the trypanosomal system where no KO conditions were employed, we also added this reference on page 7, lines 176-177.

2.) The authors have used several knockout cell lines. As they have sequenced them, it would be good to provide the mutations resulting in the loss of the protein. In my opinion it is always good to provide a corresponding rescue cell line to prove the specificity of the phenotype associated with the knockout.

We followed the suggestion and added rescue experiments for all the knockout cell lines used in the study: mS37 KO, MRM3 KO and GTPBP10 KO. The data is now shown in Fig. 3D and E; Extended Data Fig. 13B, C and 14A. It has confirmed the specificity of the described knockout phenotypes. A corresponding section was added to Methods, and the guide RNA targets are illustrated together with the sequencing results validating the knock-outs in Extended Data Fig. 13A. TRMT2B KO has previously been validated, as described in ref 15.

3.) It is not clear how the authors have generated the GTPBP10 KO cell line. Page 12/line 265 “as previously described”: there is no associated reference. In the main text they refer to Cheng et al. (2021); however, in this study a GTPBP10 KO was not generated, only a FLAG-tagged version of GTPBP10 was described. Thus, the authors should clarify this.

Thank you for noticing this mistake - we replaced the incorrect reference in the Methods section with Lavdovskaia, E. et al. The human OBG protein GTPBP10 is involved in mitoribosomal biogenesis. *Nucleic Acids Research* 46, 8471–8482 (2018).

4.) I was wondering why they could solve specifically the SSU with bound COX1 mRNA from MRM3 KO. Does it mean that COX1 mRNA is more abundant/stable than others in MRM3 KO? Why would this be the case? Any explanation or speculation?

Indeed, the structural data would suggest that the complex with COX1 mRNA is more stable than others, and we now added this information to the figure legend in Extended Data Fig. 15, and also produced Supplementary Movie 4 to show the experimental data more in detail. Unlike other leaderless mt-mRNAs, COX1 mRNA has a 5'UTR comprising three nucleotides, which would provide it with an extra stability on the mitoribosome. From the data, we could resolve that the three 5' UTR residues in the E-site are stacked with their bases against each other and against Gly164 and Gly165 of uS7m. A configuration of one or two UTR residues would not stack with uS7m in this region.

Minor points:

Lines 40-41: messenger (mRNA) and transfer (tRNA) > messenger RNA (mRNA) and transfer RNA (tRNA)

Corrected.

Line 51: ribosomal (rRNA) > ribosomal RNA (rRNA)

Corrected.

Lines 222-223, “ mS37...is lacking when the LSU is not fully assembled,...”: I think this is not accurate as there is still mS37 detectable in MRM3 KO and GTPBP10 KO gradients (Ext.

Data Fig. 13b). Better to use “reduced” or similar terms.
Corrected.

Referee #3:

Specific comments

1) The intermediates are captured by depleting TRMT2B or MRM3. Can the authors comment on how well these strategies block assembly in physiological states? In other words, how do the intermediates identified here depend on the depletion of specific components?

Thank you for highlighting this important issue, which has now been addressed in the revised manuscript. TRMT2B was chosen due to its proposed role in an earlier assembly stage, viable cell line upon depletion, and stable rRNA. This information is now added on page 2, lines 59-61, together with an additional reference to a previous work: “*depleted the enzyme TRMT2B responsible for the formation of 5-methyluridine (15,16) (see Methods), which produced a viable cell line with stable rRNA and allowed to purify intermediate preSSU particles in different states*”.

We reasoned that knock-out of MRM3 (which is an mtLSU assembly factor) would trap mtSSU maturation in the late assembly stages, and now further clarified this strategy in the text on page 5, lines 106-111: “*To explore how the SSU progresses through the mitoribosomal assembly pathway after TFB1M maturation, we reasoned that an inhibition of the corresponding LSU assembly would allow the SSU to proceed towards later stages up to the subunit joining. Therefore, next we knocked out the enzyme mitochondrial rRNA methyltransferase 3 (MRM3) responsible for a late assembly of the LSU (21), and collected cryo-EM data of the accumulated SSU and LSU particles*”.

Since overlapping assembly intermediates were detected in both models, we also clarified it on page 5, lines 111-114: “*~7% of the SSU particles in this cryo-EM data showed identical structural features and presence of the same additional density that were also detected in the particles from the TRMT2B-depleted cells, and the latter was identified as the N4-methylcytidine methyltransferase METTL15*”.

Finally, our biochemical analysis of MRM3 and GTPBP10 knockout models showed reduction of mS37 and accumulation of RBFA in both cases, suggesting that the phenotype is general, and not a factor-specific response, which strengthens the results.

2) The authors identify IF3 as an assembly factor (it would be better to name it mtIF3). What happens to mitoribosome assembly if IF3 is deleted? Is IF3 really an assembly factor or rather together with mS37 characterizes an earlier status of the initiation complex? A previous paper from the authors (PMID: 32522994) identifies already the interaction between mS37 and mtIF3 as an initial pre-initiation complex. What is the novelty here? Are the authors proposing now a different model?

As requested, “IF3” has been corrected to “mtIF3” throughout the manuscript and in the figures.

To address the comment, we have now generated a human mtIF3 knockout cell line, and the analysis is given below and in Supplementary Material. Although a monosome is not majorly affected in mtIF3 knockout (panel B), as previously observed for murine mtIF3 knockout model in ref 27, there is a mild reduction of mS37 steady-state levels (panel C) (n=3). This is consistent with the finding that mtIF3 binds before mS37, and we added this information on page 7, line 171.

As suggested, we also toned down the claims regarding mtIF3, particularly removed “repurposed” on page 10, line 224, and added a statement “additional transient intermediates are likely to exist” on page 10, lines 225-226.

The novelty of the current study compared to the mentioned work is the finding that mtIF3 binds before mS37. This information is now added in the Supplementary Movie 4. The process of the assembly/initiation appears to be generally more continuous in mitochondria, as also in trypanosomes, initiation factors are present during the assembly, and we now added the citation on page 7, line 177. We hope the revised version of the text better conveys the structural data.

3) Fig. 3d-f: How many times have these experiments been repeated?

All biochemical experiments were repeated at least three times. We have now included this information in the figure legends (Fig 3 and Extended Data Fig 13), and the data is now in the Supplementary Material.

Reviewer Reports on the First Revision:

Referee #1:

I have no further comments; the authors have responded adequately to my comments. Congratulations on such beautiful work.

Referee #2:

The authors have addressed all issues. They included the requested rescue experiments and sequence analyses for the used knockout cell lines. The flow and clarity of the text has been improved, and the provided movies are very helpful. In my opinion the revised manuscript is suitable for publication in Nature.

I only realized a small typo in Fig. 3 and ED Fig. 14: “uL12m” should be “bL12m”.

Referee #3:

The authors have addressed all raised points in this revised version, by adding new experiments including important controls. The manuscript has also an improved flow, which helps non-specialized readers to follow the findings. The authors state that biochemical experiments have been repeated three times, and show one representative blot. No statistical analysis is included.

The results of this study shed light on the biogenesis of the small subunit of the mitoribosome and the role of mS37.